# Whole-Body Conditioned Egocentric Video Prediction

**Yutong Bai**[* 1]       **Danny Tran**[* 1]       **Amir Bar**[* 2]

**Yann LeCun**[† 2,3]       **Trevor Darrell**[† 1]       **Jitendra Malik**[† 1,2]

[1]UC Berkeley (BAIR)       [2]FAIR, Meta       [3]New York University

## Abstract

We train models to **P**redict **E**go-centric **V**ideo from human **A**ctions (**PEVA**), given the past video and an action represented by the relative 3D body pose. By conditioning on kinematic pose trajectories, structured by the joint hierarchy of the body, our model learns to simulate how physical human actions shape the environment from a first-person point of view. We train an auto-regressive conditional diffusion transformer on Nymeria, a large-scale dataset of real-world egocentric video and body pose capture. We further design a hierarchical evaluation protocol with increasingly challenging tasks, enabling a comprehensive analysis of the model's embodied prediction and control abilities. Our work represents an initial attempt to tackle the challenges of modeling complex real-world environments and embodied agent behaviors with video prediction from the perspective of a human.[1]

## 1   Introduction

Human movement is rich, continuous, and physically grounded (Rosenhahn et al., 2008; Aggarwal and Cai, 1999). The way we walk, lean, turn, or reach—often subtle and coordinated—directly shapes what we see from a first-person perspective. For embodied agents to simulate and plan like humans, they must not only predict future observations (Von Helmholtz, 1925), but also understand how visual input arises from whole-body action (Craik, 1943). This understanding is essential because many aspects of the environment are not immediately visible–we need to move our bodies to reveal new information and achieve our goals.

Vision serves as a natural signal for long-term planning (LeCun, 2022; Hafner et al., 2023; Ebert et al., 2018; Ma et al., 2022). We look at our environment to plan and act, using our egocentric view as a predictive goal (Sridhar et al., 2024; Bar et al., 2025). When we consider our body movements, we should consider both actions of the feet (locomotion and navigation) and the actions of the hand (manipulation), or more generally, whole-body control (Nvidia et al., 2025; Cheng et al., 2024; He et al., 2024b; Radosavovic et al., 2024; He et al., 2024a; Hansen et al., 2024). For example, when reaching for an object, we must anticipate how our arm movement will affect what we see, even before the object comes into view. This ability to plan based on partial visual information is crucial for embodied agents to operate effectively in real-world environments.

Building a model that can effectively learn from and predict based on whole-body motion presents several fundamental challenges. First, representing human actions requires capturing both global body dynamics and fine-grained joint articulations, which involves high-dimensional, structured data with complex temporal dependencies. Second, the relationship between body movements and

---

* Equal contribution; † Equal advising.

[1]Project page: `https://dannytran123.github.io/PEVA`.

39th Conference on Neural Information Processing Systems (NeurIPS 2025).

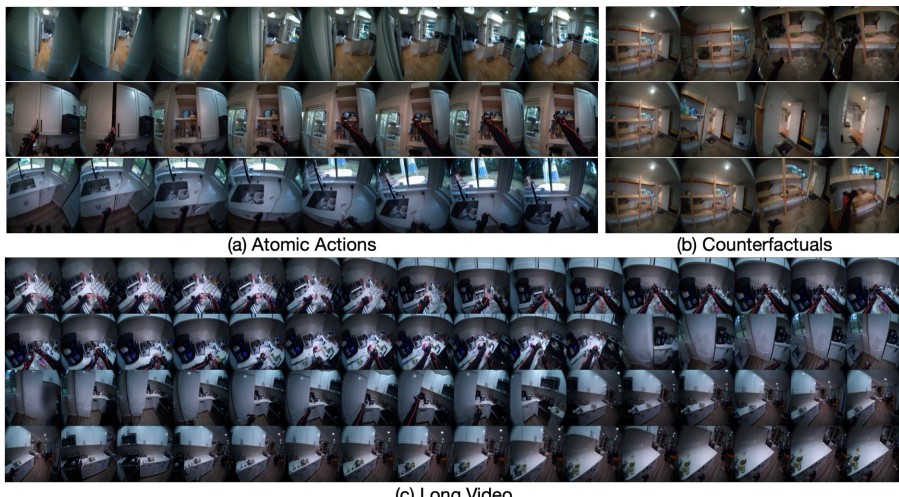

Figure 1: **Predicting Ego-centric Video from human Actions (PEVA)**. Given past video frames and an action specifying a desired change in 3D pose, PEVA predicts the next video frame. Our results show that, given the first frame and a sequence of actions, our model can generate videos of atomic actions (a), simulate counterfactuals (b), and support long video generation (c).

visual perception is highly nonlinear and context-dependent—the same arm movement can result in different visual outcomes depending on the environment and the agent's current state. Third, learning these relationships from real-world data is particularly challenging due to the inherent variability in human motion and the subtle, often delayed visual consequences of actions.

To address these challenges, we develop a novel approach **PEVA** that combines several key innovations. First, we design a structured action representation that preserves both global body dynamics and local joint movements, using a hierarchical encoding that captures the kinematic tree structure of human motion. This representation enables the model to understand both the overall body movement and the fine-grained control of individual joints. Second, we develop a novel architecture based on conditional diffusion transformers that can effectively model the complex, nonlinear relationship between body movements and visual outcomes. The architecture incorporates temporal attention mechanisms to capture long-range dependencies and a specialized action embedding component that maintains the structured nature of human motion. Third, we leverage a large-scale dataset of synchronized egocentric video and motion capture data (Ma et al., 2024), which provides the necessary training signal to learn these complex relationships. Our training strategy includes random timeskips to handle the delayed visual consequences of actions and sequence-level training to maintain temporal coherence.

For evaluation, we design a hierarchical evaluation protocol to understand PEVA's capabilities across different levels of complexity. First, we assess its ability to predict immediate visual consequences through single-step predictions. Second, we decompose complex human movements into atomic actions to test the model's understanding of how specific joint-level movements affect the egocentric view. Third, we examine the model's capability to predict long-term visual consequences by evaluating on extended time horizons, where the effects of actions may be delayed or not immediately visible. Finally, we explore the model's ability to serve as a world model for planning by using it to simulate actions and choose the ones that lead to a predefined goal. This layered approach allows us to systematically analyze the strengths and limitations of our model, revealing both its capacity to simulate embodied perception and the open challenges that remain in bridging the gap between physical action and visual experience.

To conclude, we introduce PEVA, a diffusion-based model that predicts future egocentric video conditioned on whole-body motion. By grounding prediction in 3D whole-body movement, our model captures the intricate relationship between movement and visual perception. Our comprehensive evaluation framework demonstrates that whole-body control significantly improves video quality, semantic consistency, and simulating counterfactuals.

## 2  Related Works

**World Models.** The concept of a "world model", an internal representation of the world used for prediction and planning, has a rich history across multiple disciplines. The idea was first proposed in psychology by Craik (1943), who hypothesized that the brain uses "small-scale models" of reality to anticipate events. This principle found parallel development in control theory, where methods like the Kalman Filter and Linear Quadratic Regulator (LQR) rely on an explicit model of the system to be controlled (Kalman, 1960). The idea of internal models became central to computational neuroscience for explaining motor control, with researchers proposing that the brain plans and executes movements by simulating them first (Jordan, 1996; Kawato et al., 1987; Kawato, 1999).

With the rise of deep learning, the focus shifted to learning these predictive models directly from data. Early work in computer vision demonstrated that models could learn intuitive physics from visual data to solve simple control tasks like playing billiards or poking objects (Fragkiadaki et al., 2015; Agrawal et al., 2016). This paved the way for modern, large-scale world models that predict future video frames conditioned on actions, enabling planning by "imagining" future outcomes (Ha and Schmidhuber, 2018; Hafner et al.; Liu et al., 2024; Li et al., 2022; Zhou et al., 2024; Yang et al., 2023, 2024; Assran et al., 2025). In reinforcement learning, models like Dreamer have shown that learning a world model improves sample efficiency (Hafner et al., 2023). Recent approaches have used diffusion models for more expressive generation; for example, DIAMOND generates multi-step rollouts via autoregressive diffusion (Alonso et al., 2024). In the egocentric domain, Navigation World Models (NWM) used conditional diffusion transformers (CDiT) to predict future frames from a planned trajectory (Bar et al., 2025). However, these models use low-dimensional controls and neglect the agent's own body dynamics. We build on this extensive line of work by conditioning video prediction on whole-body pose, enabling a more physically-grounded simulation.

**Human Motion Generation and Controllable Prediction.** Human motion modeling has advanced from recurrent and VAE-based methods (Rempe et al., 2021; Petrovich et al., 2021; Ye et al., 2023) to powerful diffusion-based generators (Tevet et al., 2022; Zhang et al., 2024). These models generate diverse, realistic 3D pose sequences conditioned on text (Hong et al., 2024; Guo et al., 2022; Dabral et al., 2023), audio (Ng et al., 2024; Dabral et al., 2023; Ao et al., 2023), and head pose (Li et al., 2023; Castillo et al., 2023; Yi et al., 2025). Recent works like Animate Anyone (Hu et al., 2023) and MagicAnimate (Xu et al., 2023) generate high-fidelity human animations from a reference image and pose sequence. Physically-aware extensions like PhysDiff (Yuan et al., 2023) incorporate contact into the denoising loop. While prior works treat pose as the target, our model uses it as input for egocentric video prediction, reversing the typical motion generation setup. This enables fine-grained visual control, bridging pose-conditioned video generation (Wu et al., 2023; Zhang et al., 2023) with embodied simulation. Unlike Make-a-Video (Singer et al., 2022) or Tune-A-Video (Wu et al., 2023), which focus on text/image prompts, we condition directly on physically realizable body motion.

**Egocentric Perception and Embodied Forecasting.** Egocentric video datasets such as Ego4D (Grauman et al., 2022), Ego-Exo4D (Grauman et al., 2024) and EPIC-KITCHENS (Damen et al., 2018) were used to study human action recognition, object anticipation (Furnari and Farinella, 2020), future video prediction (Girdhar and Grauman, 2021), and even animal behavior (Bar et al., 2024). To study pose estimation, EgoBody (Zhang et al., 2022) and Nymeria (Ma et al., 2024) provide synchronized egocentric video and 3D pose. Unlike these works, we treat future body motion as a control signal, enabling visually grounded rollout. Prior works in egocentric pose forecasting (Yuan and Kitani, 2019) and visual foresight (Finn and Levine, 2017) show that predicting future perception supports downstream planning. Our model unifies these lines by predicting future egocentric video from detailed whole-body control, enabling first-person planning with physical and visual realism.

## 3  PEVA

In this section we describe our whole-body-conditioned ego-centric video prediction model. We start by describing how to represent human actions (Section 3.1), then move on to describe the model and the training objective (Section 3.2). Finally, we describe the model architecture in Section 3.3.

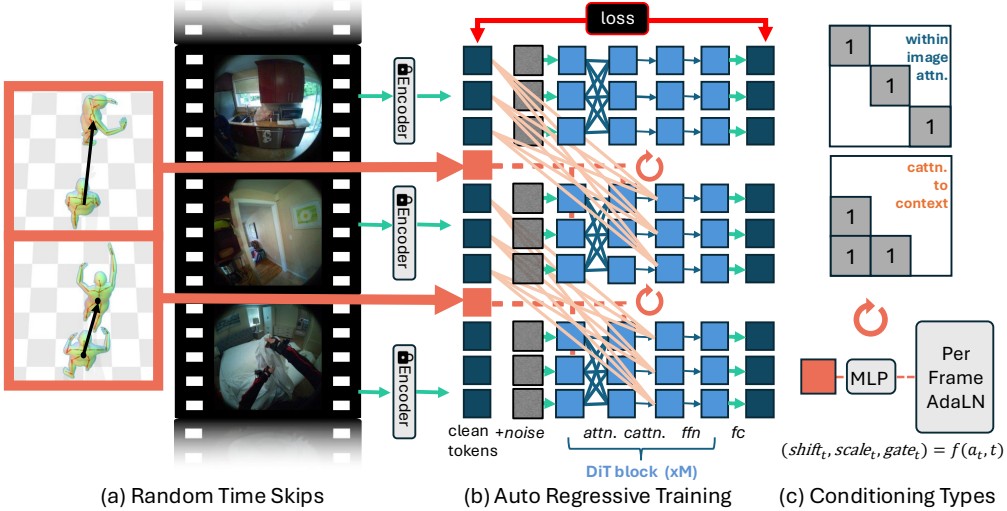

| (a) Random Time Skips | (b) Auto Regressive Training | (c) Conditioning Types |

Figure 2: **Design of PEVA.** To train on an input video, we choose a random subset of frames and encode them via a fixed encoder (a). They are then fed to a CDiT that is trained autoregressively with teacher forcing (b). During the denoising process, each token attends to same-image tokens and cross-attends to clean tokens from past image(s). Action conditioning is done via AdaLN layers.

## 3.1   Structured Action Representation from Motion Data

To effectively capture the relationship between human motion and egocentric visual perception, we define each action as a high-dimensional vector encoding both global body dynamics (global translation via the root joint) and detailed joint articulations structured by the kinematic tree. We synchronize motion capture data with egocentric video frames using timestamps and convert global coordinates into a local frame centered at the pelvis, ensuring invariance to initial position and orientation. Global positions are converted to local coordinates, quaternions to relative Euler angles, and joint relationships are preserved using the kinematic hierarchy.  We normalize all motion parameters for stable learning: positions are scaled to $[-1, 1]$ and rotations bounded within $[-\pi, \pi]$. Each action reflects the change between consecutive frames, allowing the model to learn how physical movements produce visual outcomes. Our motion representation follows the Xsens skeleton (Movella, 2021; Ma et al., 2024), which shares the kinematic tree of SMPL (Loper et al., 2023) but differs in joint set and ordering, and omits body shape parameters.

## 3.2   Ego-Centric Video Prediction for Whole-Body Control

Next, we describe our formulation of **PEVA** from the perspective of an embodied agent. Intuitively, the model is an autoregressive diffusion model that receives an input video and a corresponding sequence of actions describing how the agent moves and acts. Given any prefix of frames and actions, the model predicts the resulting state of the world after applying the last action and considering other environment dynamics.

More formally, we are given a dataset $D = \{(x_0, a_0, ..., x_T, a_T)\}_{i=1}^{n}$ of agents videos from egocentric view and their associated body controls, such that every $x_j \in \mathbb{R}^{H \times W \times 3}$ is a video frame and $a_j \in \mathbb{R}^{d_{act}}$ an action in the Xsens skeleton ordering (Movella, 2021) for the upper body (everything above the pelvis), representing the change in translation, together with the delta rotation of all joints relative to the previous joint rotation. We represent motion in 3D space, thus we have 3 degrees of freedom for root translation, 23 joints for the body and represent relative joint rotations as Euler angles in 3D space leaving $d_{act} = 3 + 23 \times 3 = 72$.

We start by encoding each individual frame $s_i = \text{enc}(x_i)$ into a corresponding state representation, through a pre-trained VAE encoder (Rombach et al., 2022). Given a sequence of controls $a_0, \ldots a_T$, our goal is to build a generative model that captures the dynamics of the environment:

$$P(s_T, \ldots s_0 | a_T, \ldots a_0) = P(s_0) \prod_{t=0}^{T-1} P(s_{t+1} | s_t, \ldots, s_0, a_T, \ldots a_0) \qquad (1)$$

To simplify the model, we factorize the distribution and make a Markov assumption that the next state is dependent on the last $k$ states and a single past action:

$$P(s_{t+1}|s_t, \ldots, s_0, a_T, \ldots a_0) = P(s_{t+1}|s_t, \ldots, s_{t-k+1}, a_{t-1}) \tag{2}$$

We aim to train a model parametrized by $\theta$ that minimizes the negative log-likelihood:

$$\hat{\theta} = \arg\min_\theta \left[ -\log P_\theta(s_0) - \sum_{t=0}^{T-1} \log P_\theta(s_{t+1}|s_t, \ldots, s_{t-k+1}, a_t) \right]$$

We model each transition $P_\theta(s_{t+1}|s_t, \ldots, s_{t-k+1}, a_t)$ using a Denoising Diffusion Probabilistic Model (DDPM) (Ho et al., 2020), which maximizes the (reweighted) evidence lower bound (ELBO) of the log-likelihood. For each transition, we define the forward diffusion process $q(z_\tau \mid s_{t+1}) = \mathcal{N}(z_\tau; \sqrt{\bar{\alpha}_\tau} s_{t+1}, (1 - \bar{\alpha}_\tau)\mathbf{I})$, where $z_\tau$ is the noisy version of $s_{t+1}$ at noise timestep $\tau$, and $\bar{\alpha}_\tau$ is the cumulative product of noise scales. The reverse process is learned by training a neural network $\epsilon_\theta$ to predict the noise given $z_\tau$ and the conditioning context $c_t = (s_t, \ldots, s_{t-k+1}, a_t)$.

Then denoising loss term for a transition is given by:

$$\mathcal{L}_{\text{simple, t}} = \mathbb{E}_{\tau, \epsilon \sim \mathcal{N}(0,I)} \left[ \left\| \epsilon - \epsilon_\theta \left( \sqrt{\bar{\alpha}_\tau} s_{t+1} + \sqrt{1 - \bar{\alpha}_\tau}\epsilon, \ c_t, \ \tau \right) \right\|^2 \right] \tag{3}$$

Where $\mathcal{L}_{\text{simple, 0}}$ is the loss term corresponding to the unconditional generation of $s_0$. Additionally, we also predict the covariances of the noise, and supervise them using the full variational lower bound loss $\mathcal{L}_{\text{vlb,t}}$ as proposed by (Nichol and Dhariwal, 2021).

Hence the final objective yields a (weighted) version of the ELBO for each term in the sequence:

$$\mathcal{L} = \sum_{t=0}^{T-1} \mathcal{L}_{simple,t} + \lambda \mathcal{L}_{vlb,t} \tag{4}$$

Despite not being a lower bound of the log-likelihood, the reweighted ELBO works well in practice for image generation with transformers (Nichol and Dhariwal, 2021; Peebles and Xie, 2023).

The advantage of our formulation is that it allows training in parallelized fashion using causal masking. Given a sequence of frames and actions, we can train on every prefix of the sequence in a single forward-backward pass. Next, we elaborate on the architecture of our model.

### 3.3 Autoregressive Conditional Diffusion Transformer

While prior work in navigation world models (Bar et al., 2025) focuses on simple control signals like velocity and heading, modeling whole-body human motion presacents significantly greater challenges. Human activities involve complex, coordinated movements across multiple degrees of freedom, with actions that are both temporally extended and physically constrained. This complexity necessitates architectural innovations beyond standard CDiT approaches.

To address these challenges, we extend the Conditional Diffusion Transformer (CDiT) architecture with several key modifications that enable effective modeling of whole-body motion:

**Random Timeskips.** Human activities often span long time horizons with actions that can take several seconds to complete. At the same time, videos are a raw signal which requires vast amounts of compute to process. To handle video more efficiently, we introduce random timeskips during training (see Figure 2a), and include the timeskip as an action to inform the model's prediction. This allows the model to learn both short-term motion dynamics and longer-term activity patterns. Learning long-term dynamics is particularly important for modeling activities like reaching, bending, or walking, where the full motion unfolds over multiple seconds. In practice, we sample 16 video frames from a 32 second window.

**Sequence-Level Training.** Unlike NWM which predicts single frames, we model the entire sequence of motion by applying the loss over each prefix of frames following Eq. 4. We include an example of this in Figure 2b. This is crucial because human activities exhibit strong temporal dependencies - the way someone moves their arm depends on their previous posture and motion. We enable efficient training by parallelizing across sequence prefixes through spatial-only attention in the current frame

and past-frame-only attention for historical context (Figure 2c). In practice we train models with sequences of 16 frames.

**Action Embeddings.** The high-dimensional nature of whole-body motion (joint positions, rotations, velocities) requires careful handling of the action space. We take the most simple strategy: we concatenate all actions in time $t$ into a $1D$ tensor which is fed to each AdaLN layer for conditioning (see Figure 2c).

These architectural innovations are essential for modeling the rich dynamics of human motion. By training on sequence prefixes and incorporating timeskips, our model learns to generate temporally coherent motion sequences that respect both short-term dynamics and longer-term activity patterns. The specialized action embeddings further enable precise control over the full range of human movement, from subtle adjustments to complex coordinated actions.

### 3.4 Inference and Planning with PEVA

**Sampling procedure at test time.** Given a set of context frames $(x_t, ..., x_{t-k+1})$, we encode these frames to get $(s_t, ..., s_{t-k+1})$ and pass the encoded context as the clean tokens in Figure 2b and pass in randomly sampled noise as the last frame. We then follow the DPPM sampling process to denoise the last frame conditioning on our action. For faster inference time, we employ special attention masks where we change the mask in Figure 2c for within image attention to only be applied on the tokens of the last frame and change the mask for cross attention to context so that cross attention is only applied for the last frame.

**Autoregressive rollout strategy.** To follow a set of actions we use an autoregressive rollout strategy. Given an initial set of context frames we $(x_t, ..., x_{t-k+1})$ we start by encoding each individual frame to get $(s_t, ..., s_{t-k+1})$ and add the current action to create the conditioning context $c_t = (s_t, ..., s_{t-k+1}, a_t)$. We then sample from our model parameterized by $\theta$ to generate the next state: $s_{t+1} = P_\theta(s_{t+1}|c_t)$. We then discard the first encoding and append the generated $s_{t+1}$ and add the next action to produce the next context $c_{t+1} = (s_{t+1}, s_t, ..., s_{t-k+1}, a_{t+1})$. We then repeat the process for our entire set of actions. Finally, to visualize the predictions, we decode the latent states to pixels using the VAE decoder (Rombach et al., 2022).

## 4 Experiments and Results

### 4.1 Experiment Setting

**Dataset.** We use the Nymeria dataset (Ma et al., 2024), which contains synchronized egocentric video and full-body motion capture, recorded in diverse real-world settings using an XSens system (Movella, 2021). Each sequence includes RGB frames and 3D body poses in the XSens skeleton format, covering global translation and rotations of body joints. We sample body motions at 4 FPS. Videos are center-cropped and resized to $224{\times}224$. We split the dataset 80/20 for training and evaluation, and report all metrics on the validation set.

**Training Details.** We train variants of Conditional Diffusion Transformer (CDiT-S to CDiT-XXL, up to 32 layers) using a context window of 3–15 frames and predicting 64-frame trajectories. Models operate on $2{\times}2$ patches and are conditioned on both pose and temporal embeddings. We use AdamW (lr=8e$-5$, betas=(0.9, 0.95), grad clip=10.0) and batch size 512. Action inputs are normalized to $[-1, 1]$ for translation and $[-\pi, \pi]$ for rotation. All experiments use Stable Diffusion VAE tokenizer and follow NWM's hardware and evaluation setup. Metrics are averaged over 5 samples per sequence.

### 4.2 Comparison with Baselines

To comprehensively evaluate our model, we compare PEVA with CDiT (Bar et al., 2025) and Diffusion Forcing (Chen et al., 2024) along two key dimensions. First, to assess whether the model faithfully simulates future observations conditioned on actions, we evaluate the perceptual and semantic similarity, and action consistency of the generated frames. We use LPIPS (Zhang et al., 2018a) and DreamSim (Fu et al., 2023), to measure perceptual and semantic similarity. To assess action consistency, we evaluate how faithfully the model follows the intended actions by measuring camera motion, using Absolute Trajectory Error (ATE) and Relative Pose Error (RPE) (Sturm et al., 2012), and 2D wrist keypoints, using Percentage of Correct Keypoints (PCK@0.2), precision,

recall, and accuracy. More details on the camera pose estimation and wrist position evaluation are in Supplementary Section 7. Second, to evaluate the overall quality and realism of the generated samples, we report FID (Heusel et al., 2017).

As shown in Table 1, our model achieves better results on both action consistency and generative quality. Furthermore, Figure 3 shows that our models tend to maintain lower FID scores than the baselines as the prediction horizon increases, suggesting improved visual quality and temporal consistency over longer rollouts. Qualitative results for 16 second rollouts can be seen in Figure 1c, Figure 5, and Supplementary Figures 22– 34. We implement Diffusion Forcing (DF*) on top of PEVA by applying the diffusion forward process to the entire sequence of encoded latents, then predicting the next state given the previous (noisy) latents. At test time, we autoregressively predict the next state as in PEVA, without injecting noise into previously predicted frames, like Chen et al. (2024).

We additionally, compare PEVA against Cosmos (Agarwal et al., 2025) on wrist position accuracy with PCK@0.2. We find that whole-body actions as a control signal leads to significantly finer control compared to text conditioning, with PEVA achieving an accuracy of 0.85 compared to 0.22 for Cosmos.

Table 1: **Baseline Metrics.** Comparison of baselines on single-step prediction 2 seconds ahead.

| Model | LPIPS ↓ | DreamSim ↓ | ATE ↓ | RPE (Trans.) ↓ | RPE (Rot.) ↓ | PCK@0.2 ↑ | Precision ↑ | Recall ↑ | Accuracy ↑ | FID ↓ |
|---|---|---|---|---|---|---|---|---|---|---|
| DF* | $0.352^{0.003}$ | $0.244^{0.003}$ | 0.464 | 0.422 | 36.514 | 0.750 | 0.914 | 0.592 | 0.679 | $73.052^{1.101}$ |
| CDiT | $0.313^{0.001}$ | $0.202^{0.002}$ | 0.386 | 0.367 | 26.057 | 0.755 | 0.865 | 0.833 | 0.794 | $63.714^{0.491}$ |
| PEVA | $0.303^{0.001}$ | $0.193^{0.002}$ | 0.274 | 0.266 | 13.527 | 0.791 | 0.923 | 0.888 | 0.871 | $62.293^{0.671}$ |

## 4.3 Atom Actions Control

To evaluate PEVA's ability to follow structured physical control, we decompose complex motions into atomic actions. We extract video segments exhibiting fundamental movements—such as hand motions (up, down, left, right) and whole-body actions (forward, rotate)—based on thresholded positional deltas. We sample 100 examples per action type, and evaluate single-step prediction 2 seconds ahead. Qualitative results are shown in Figure 1a and Figure 4, and quantitative results in Table 2. See Supplementary Figure 21 for more qualitative examples of interaction rich atomic actions.

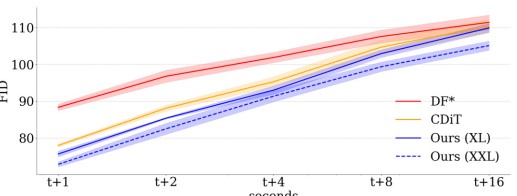

Figure 3: **Video Quality Across Time (FID).** Comparison of generation accuracy and quality as a function of time (up to 16s). Qualitative results for 16-second rollouts can be seen in Figure 1c, Figure 5, and Supplementary Figures 22– 34.

Table 2: **Atomic Action Performance.** Comparison of models in generating videos of atomic actions evaluated on LPIPS.

| Model | Navigation | | | Left Hand | | | | Right Hand | | | |
|---|---|---|---|---|---|---|---|---|---|---|---|
| | Forward | Rot.L | Rot.R | Left | Right | Up | Down | Left | Right | Up | Down |
| DF* | $0.393^{0.011}$ | $0.314^{0.006}$ | $0.279^{0.005}$ | $0.292^{0.009}$ | $0.306^{0.005}$ | $0.332^{0.008}$ | $0.323^{0.006}$ | $0.304^{0.006}$ | $0.315^{0.007}$ | $0.305^{0.005}$ | $0.296^{0.008}$ |
| CDiT | $0.348^{0.004}$ | $0.284^{0.003}$ | $0.249^{0.004}$ | $0.258^{0.005}$ | $0.265^{0.009}$ | $0.279^{0.008}$ | $0.267^{0.004}$ | $0.286^{0.007}$ | $0.273^{0.004}$ | $0.277^{0.004}$ | $0.268^{0.002}$ |
| Ours (XL) | $0.337^{0.006}$ | $0.277^{0.006}$ | $0.242^{0.007}$ | $0.244^{0.005}$ | $0.257^{0.004}$ | $0.272^{0.008}$ | $0.263^{0.003}$ | $0.271^{0.005}$ | $0.267^{0.003}$ | $0.268^{0.004}$ | $0.256^{0.009}$ |
| Ours (XXL) | $0.325^{0.006}$ | $0.269^{0.005}$ | $0.234^{0.004}$ | $0.236^{0.003}$ | $0.241^{0.003}$ | $0.251^{0.004}$ | $0.247^{0.005}$ | $0.256^{0.007}$ | $0.254^{0.005}$ | $0.252^{0.004}$ | $0.245^{0.005}$ |

## 4.4 Ablation Studies

We first conduct ablation studies to assess the impact of context length, action representation, and model size in PEVA summarized in Table 3. First, increasing the context window from 3 to 15 frames consistently improves performance across all metrics, highlighting the importance of temporal context for egocentric prediction. Second, we compare two action embedding strategies—MLP-based encoding versus simple concatenation—and find that the latter performs competitively despite its simplicity, suggesting that our structured action representation already captures sufficient motion information. Third, model scale plays a significant role: larger variants from PEVA-S to PEVA-XXL show steady gains in perceptual and semantic fidelity. The gray-highlighted row denotes the default configuration in main experiments. See Supplementary Sections 8.1, 8.2, and 8.3 for additional ablations on action type, timeskips, and action conditioning.

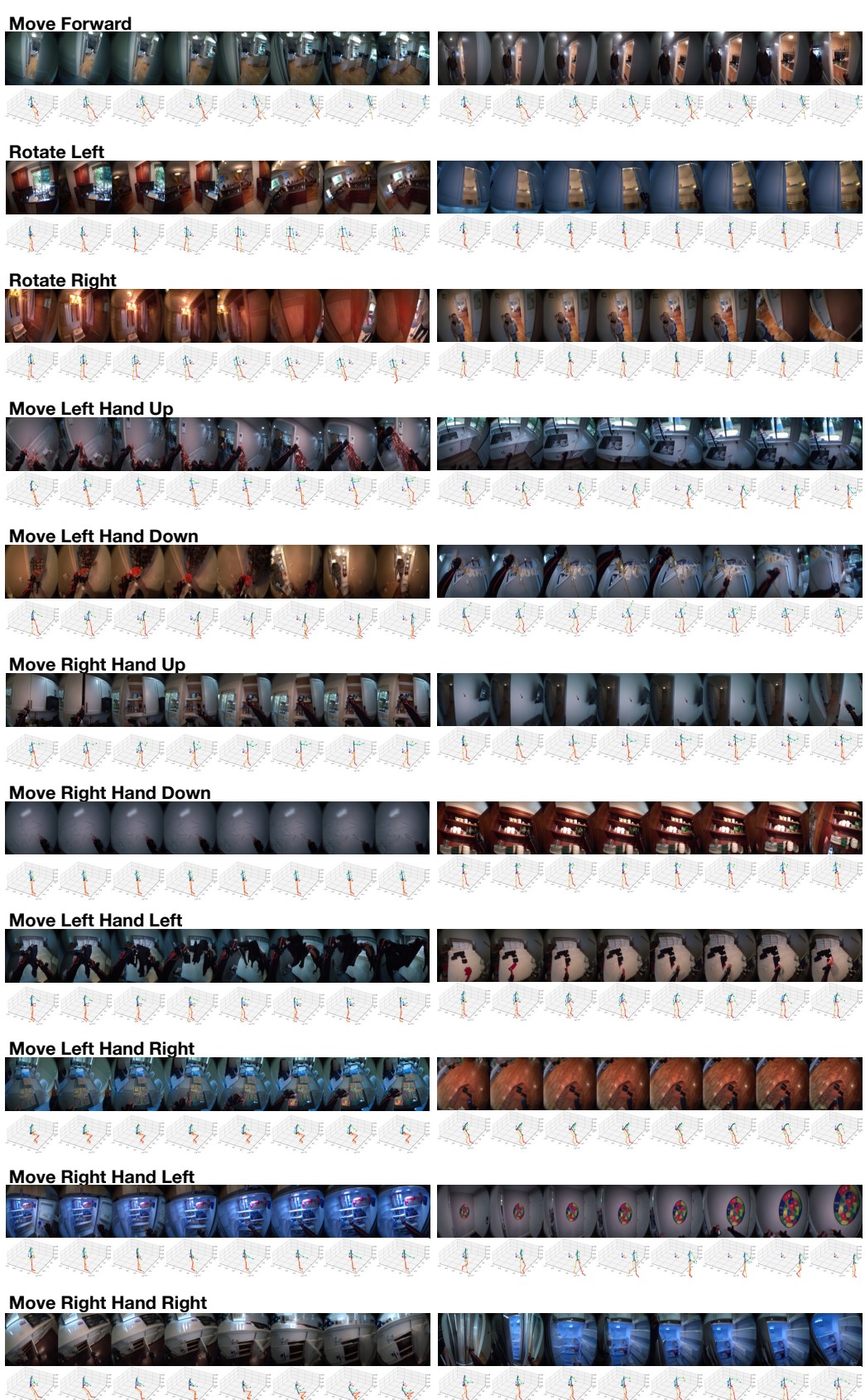

Figure 4: **Atom Actions Generation**. We include video generation examples of different atomic actions specified by 3D-body poses.

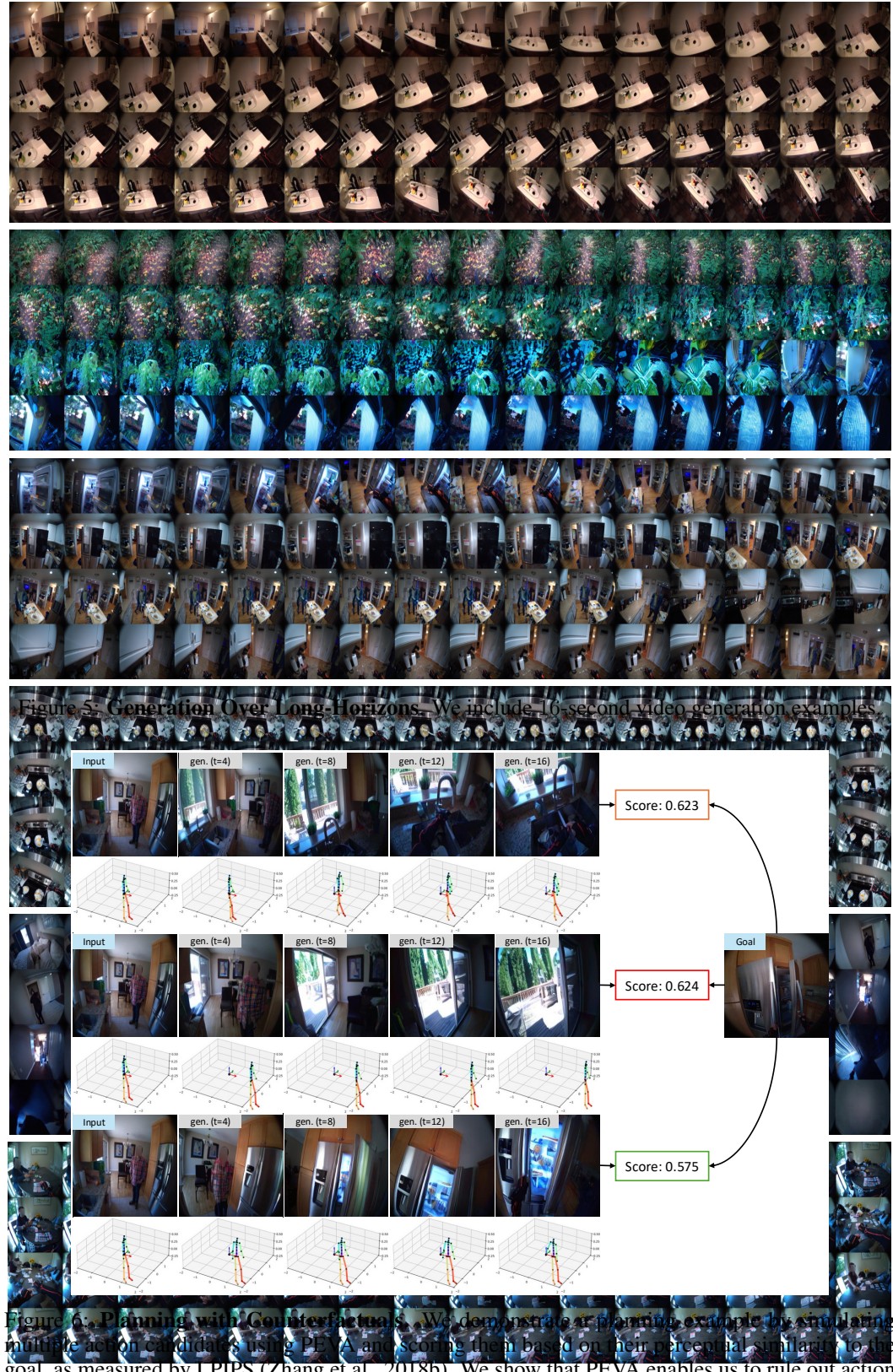

Figure 5: **Generation Over Long Horizons.** We include 16-second video generation examples.

Figure 6: **Planning with Counterfactuals.** We demonstrate a planning example by simulating multiple action candidates using PEVA and scoring them based on their perceptual similarity to the goal, as measured by LPIPS (Zhang et al., 2018b). We show that PEVA enables us to rule out action sequences that leads us to the sink in the top row, and outdoors in the second row.

Table 3: **Model Ablations**. We evaluate the impact of different context lengths, action embedding methods, and model sizes on single-step prediction performance (2 seconds into the future).

| Configuration | Metrics | | | |
|---|---|---|---|---|
| | LPIPS ↓ | DreamSim ↓ | PSNR ↑ | FID ↓ |
| *Context Length* | | | | |
| 3 frames | $0.304^{0.002}$ | $0.199^{0.003}$ | $16.469^{0.044}$ | $63.966^{0.421}$ |
| 7 frames | $0.304^{0.001}$ | $0.195^{0.002}$ | $16.443^{0.068}$ | $62.540^{0.314}$ |
| 15 frames | $0.303^{0.001}$ | $0.193^{0.002}$ | $16.511^{0.061}$ | $62.293^{0.671}$ |
| *Action Representation* | | | | |
| Action Embedding ($d = 512$) | $0.317^{0.003}$ | $0.202^{0.002}$ | $16.195^{0.081}$ | $63.101^{0.341}$ |
| Action Concatenation | $0.303^{0.001}$ | $0.193^{0.002}$ | $16.511^{0.061}$ | $62.293^{0.671}$ |
| *Model Size* | | | | |
| PEVA-S | $0.370^{0.002}$ | $0.327^{0.002}$ | $15.743^{0.060}$ | $101.38^{0.450}$ |
| PEVA-B | $0.337^{0.001}$ | $0.246^{0.002}$ | $16.013^{0.091}$ | $74.338^{1.057}$ |
| PEVA-L | $0.308^{0.002}$ | $0.202^{0.001}$ | $16.417^{0.037}$ | $64.402^{0.496}$ |
| PEVA-XL | $0.303^{0.001}$ | $0.193^{0.002}$ | $16.511^{0.061}$ | $62.293^{0.671}$ |
| PEVA-XXL | $0.298^{0.002}$ | $0.186^{0.003}$ | $16.556^{0.060}$ | $61.100^{0.517}$ |

### 4.5 Long-Term Prediction Quality

We evaluate the model's ability to maintain visual and semantic consistency over extended prediction horizons. As shown in Figure 5 and Supplementary Figures 22– 34, PEVA generates coherent 16-second rollouts conditioned on full-body motion. Table 3 reports DreamSim scores at increasing time steps, showing a gradual degradation from 0.178 (1s) to 0.390 (16s), indicating that predictions remain semantically plausible even far into the future. PEVA also demonstrates physical realism over longer sequences, as seen in Supplementary Figure 14.

### 4.6 Planning with Multiple Action Candidates.

We first test localized head and hand counterfactuals in Supplementary Figure 15. We then demonstrate samples where PEVA enables planning with multiple action candidates in Figure 1b, Figure 6, and Supplementary Figures 18– 20. We start by sampling multiple action candidates and simulate each action candidate using PEVA via autoregressive rollout. Finally, we rank each action candidate's final prediction by measuring LPIPS similarity with the goal image. We find that PEVA is effective in enabling planning through simulating action candidates.

## 5 Failure Cases, Limitations and Future Directions

While PEVA shows strong results in predicting egocentric video from whole-body motion, several limitations remain. First, our planning evaluation is preliminary, we only perform simulation-based selection over single-arm actions (see Supplementary Section 9). This demonstrates early planning capability but not full trajectory optimization. Extending PEVA to closed-loop or interactive environments is an important next step. Second, the model lacks explicit conditioning on task intent or semantic goals, instead relying on image similarity. Third, long-horizon degradation arises from error accumulation in open-loop generation. We plan to address this through feedback in interactive environments, or by incorporating text-conditioned supervision supervision to anchor predictions.

## 6 Conclusion

We introduced PEVA, a diffusion-based model that predicts egocentric video from detailed 3D human motion. Unlike prior work that use low-dimensional control, PEVA leverages full-body pose sequences to simulate realistic and controllable visual outcomes. Trained on Nymeria, it captures the link between movement and egocentric perception. Experiments show improvements in prediction quality, semantic consistency, and fine-grained control over strong baselines. Our hierarchical evaluation highlights the value of whole-body conditioning across short-term, long-horizon, and atomic action tasks. While preliminary, our planning results demonstrate the potential for simulating embodied action consequences, moving toward more grounded models of perception and action.

## Acknowledgment

The authors thank Rithwik Nukala for his help in annotating atomic actions. We thank Katerina Fragkiadaki, Philipp Krähenbühl, Bharath Hariharan, Guanya Shi, Shubham Tulsiani and Deva Ramanan for the useful suggestions and feedbacks for improving the paper; Jianbo Shi for the discussion regarding control theory; Yilun Du for the support on Diffusion Forcing; Brent Yi for his help in human motion related works and Alexei Efros for the discussion and debates regarding world models. This work is partially supported by the ONR MURI N00014-21-1-2801.

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

# Supplementary Material

The structure of the Appendix is as follows: we start by providing additional details for our Camera Pose estimation evaluation and Wrist Position evaluation in Section 7, then provide additional ablation studies in Section 8, then some additional planning attempts with PEVA in Section 9, then include additional qualitative results in Section 10, and then include details about training and inference time in Section 11.

## 7   Camera Pose Estimation and Wrist Position Evaluation

We assess how faithfully PEVA follows head and hand motion using two evaluations. Camera Pose Estimation: we extract egocentric camera poses from generated frames using VGGT (Wang et al., 2025) and compare them with ground truth using Absolute Trajectory Error (ATE) and Relative Pose Error (RPE) for translation and rotation (Sturm et al., 2012). Wrist Pose Evaluation: we extract 2D wrist keypoints and compute PCK@0.2, precision, recall, and accuracy. These metrics jointly measure whether the generated videos accurately reflect the conditioned action signals, both globally (head-driven motion) and locally (hand alignment).

## 8   Additional Abalation Studies

We conduct two additional ablation studies. First, we ablate our Whole-Body actions against actions that only include the Head and Hands in Section 8.1. Second, we ablate our Random Timeskips against fixed frame rates in Section 8.2. Third, we ablate the impact of our action conditioning in Section 8.3.

### 8.1   Action Type Ablation

We conduct ablation studies on the action type. We compare our 72D whole-body actions with only the 6-DoF poses of the head, left hand, and right hand relative to their previous states leaving $d_{act} = 3 \times 6 = 18$ summarized in Table 4 for the camera pose and wrist position evaluation. Our Whole-Body actions perform better than only using the Head and Hands across almost all metrics. These results indicate that non-visible body joints, such as those in the torso and legs, meaningfully affect egocentric visual dynamics, such as through head stabilization, balance, and intent. We find that full-body conditioning improves both spatial grounding and long-term consistency.

Table 4: **Action Type Ablations.** We evaluate the impact of different Action Types on single-step prediction performance (2 seconds into the future) on the camera pose and wrist positions.

| Action Type | ATE ↓ | RPE (Translation) ↓ | RPE (Rotation) ↓ | PCK@0.2 ↑ | Precision ↑ | Recall ↑ | Accuracy ↑ |
|---|---|---|---|---|---|---|---|
| Head + Hands | 0.381 | 0.412 | 38.425 | 0.692 | 0.906 | 0.722 | 0.756 |
| Whole-Body | 0.351 | 0.346 | 26.578 | 0.877 | 0.890 | 0.907 | 0.858 |

### 8.2   Random Timeskip Ablation

We ablate our random timeskips against fixed frame rates summarized in Table 5. The results demonstrate that fixed-rate models tend to overfit to sepcific temporal patterns and degrade at longer horizons. In contrast, random timeskips encourage temporally invariant learning, improving both short and long term prediction quality.

### 8.3   Influence of Action Conditioning on Visual Prediction

We examine how the action signal influences vision prediction in Figure 7. We find that the further we deviate from the ground-truth action, the further the visual prediction is from ground truth. This demonstrates the the model meaningfully conditions on the action for visual prediction, as opposed to simply relying solely on the visual context.

Table 5: **Random Timeskips Ablation.** We evaluate the impact of our Random Timeskips compared to various fixed frame rates measured on DreamSim ↓.

| Method | 1s | 2s | 4s | 8s | 16s | Avg |
|---|---|---|---|---|---|---|
| 4 Hz | 0.175 | 0.218 | 0.279 | 0.343 | 0.434 | 0.290 |
| 2 Hz | 0.167 | 0.198 | 0.240 | 0.291 | 0.358 | 0.251 |
| 0.5 Hz | 0.175 | 0.200 | 0.235 | 0.265 | 0.319 | 0.239 |
| Random Skips | 0.169 | 0.198 | 0.236 | 0.268 | 0.293 | 0.233 |

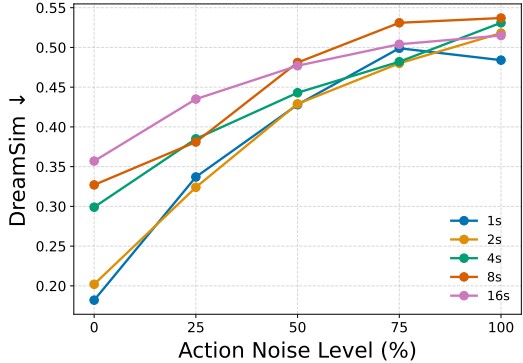

Figure 7: **Visual Quality Across Noise Levels.** Evaluation of the visual quality as we add noise the to the action. 0% is just the action whereas 100% is pure gaussian noise.

# 9 Some planning attempts with PEVA

Here we describe how to use a trained PEVA to plan action sequences to achieve a visual target. We formulate planning as an energy minimization problem and perform standalone planning in the same way as NWM (Bar et al., 2025) using the Cross-Entropy Method (CEM) (Rubinstein, 1997) besides minor modifications in the representation and initialization of the action.

For simplicity, we conduct two experiments where we only predict moving either the left or right arm controlled by predicting the relative joint rotations represented as euler angles. For each respective arm we control only the shoulder, upper arm, forearm, and hand leaving our actions space as $4 \times 3 = 12$ where we have $(\Delta\phi_{\text{shoulder}}, \Delta\theta_{\text{shoulder}}, \Delta\psi_{\text{shoulder}}, ..., \Delta\phi_{\text{forearm}}, \Delta\theta_{\text{forearm}}, \Delta\psi_{\text{forearm}})$. We initialize mean $(\mu_{\Delta\phi_{\text{shoulder}}}, \mu_{\Delta\theta_{\text{shoulder}}}, \mu_{\Delta\psi_{\text{shoulder}}}, ..., \mu_{\Delta\phi_{\text{forearm}}}, \mu_{\Delta\theta_{\text{forearm}}}, \mu_{\Delta\psi_{\text{forearm}}})$ and variance $(\sigma^2_{\Delta\phi_{\text{shoulder}}}, \sigma^2_{\Delta\theta_{\text{shoulder}}}, \sigma^2_{\Delta\psi_{\text{shoulder}}}, ..., \sigma^2_{\Delta\phi_{\text{forearm}}}, \sigma^2_{\Delta\theta_{\text{forearm}}}, \sigma^2_{\Delta\psi_{\text{forearm}}})$ as the mean and variance of the next action across the training dataset for these segments.

Table 6: Mean and Variance of relative rotation as euler angles $(\phi, \theta, \psi)$ for arm segments computed across the training dataset.

| Segment | Statistic | Right Arm | Left Arm |
|---|---|---|---|
| Shoulder | Mean | $(0.0027, -0.0012, -0.0015)$ | $(0.0624, 0.0687, 0.1494)$ |
| | Variance | $(0.0010, -0.0006, 0.0003)$ | $(0.0625, 0.0697, 0.1496)$ |
| Upper Arm | Mean | $(0.0107, -0.0011, -0.0020)$ | $(0.1119, 0.1647, 0.1791)$ |
| | Variance | $(-0.0062, -0.0004, -0.0013)$ | $(0.0991, 0.1593, 0.1611)$ |
| Forearm | Mean | $(0.0068, -0.0035, 0.0077)$ | $(0.1937, 0.2107, 0.2261)$ |
| | Variance | $(-0.0036, -0.0063, 0.0002)$ | $(0.1791, 0.2012, 0.2186)$ |
| Hand | Mean | $(0.0065, 0.0001, 0.004,)$ | $(0.2417, 0.229, 0.2631)$ |
| | Variance | $(-0.0024, -0.0032, -0.0001)$ | $(0.2126, 0.2237, 0.2475)$ |

We assume the action is a straight continuous motion. Thus we repeat this action for our sequence length, in our case $T = 8$ and optimize the delta actions. The time interval between steps is fixed at $k = 0.25$ seconds. All other hyperparameters remain the same as in NWM (Bar et al., 2025).

## 9.1 Qualitative Results

Due to time constraints, we focus our investigation on arm movements—arguably the most complex among body actions. While this remains an open problem, we present preliminary results using PEVA with CEM for standalone planning. This setting simplifies the high-dimensional control space while still capturing key challenges of full-body coordination.

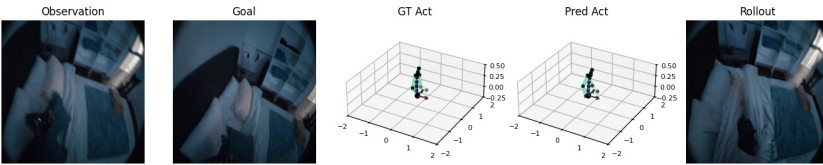

Figure 8: In this case, we are able to predict a sequence of actions that pulls our left arm in, similar to the goal.

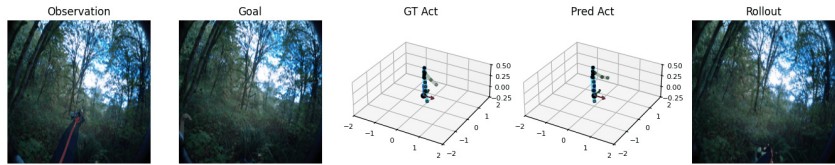

Figure 9: In this case, we are able to predict a sequence of actions that lowers our left arm, but not the same amount as the groundtruth sequence as we can see in the pose and hand at the bottom of our rollout.

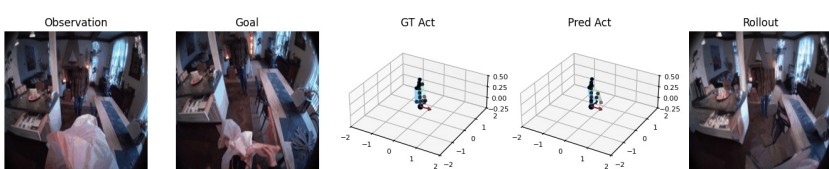

Figure 10: In this case, we are able to predict a sequence of actions that lowers our left arm that lowers the tissue. However, the goal image still has the tissue visible.

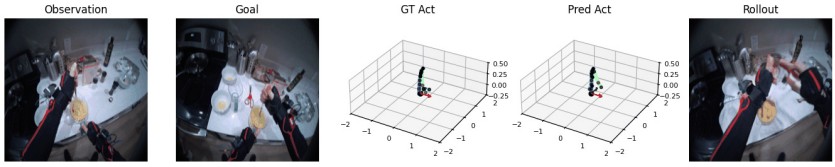

Figure 11: In this case, we are able to predict a sequence of actions that raises our right arm to the mixing stick. We see a limitation with our method as we only predict the right arm so we do not predict to move the left arm down accordingly.

## 10 More Qualitative Results

In this section, we present additional qualitative results following the same settings in the main paper. These results are organized into three parts: counterfactual simulations in Section 10.1, atomic action generation in Section 10.2, and long-horizon video generation in Section 10.3.

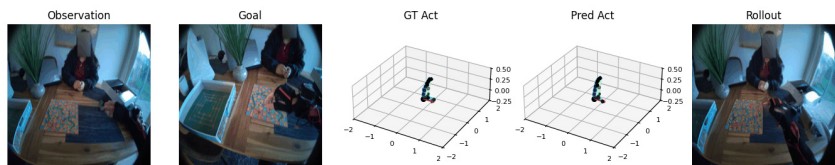

Figure 12: In this case, we are able to predict a sequence of actions that moves our right arm toward the left but not quite enough. We see a limitation with our method as we only predict the right arm so we do not predict any necessary additional body rotations.

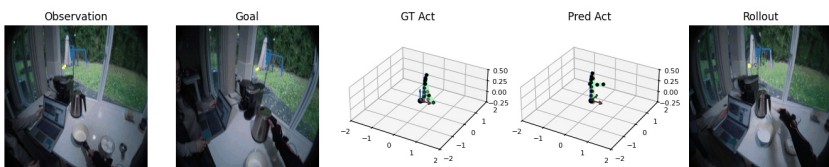

Figure 13: In this case, we are able to predict a sequence of actions that reaches toward the kettle but does not quite grab it as in the goal.

## 10.1 Additional Counterfactual Qualitative Results

In this section, we include additional counterfactual qualitative results. First, demonstrate that our model is able to capture the physical realism through a counterfactual example of continuing to pull on an open drawer and correctly follow real world physical constraints in Figure 14. Second, we include counterfactual examples of head and movement in Figure 15. Third, we include more examples of using counterfactuals for visual planning in Figures 16– 20.

## 10.2 Additional Atomic Actions

We provide additional atomic action rollouts that are more interaction rich in Figure 21.

## 10.3 Additional Long Video Generation

We provide additional examples of long video generation in Figures 22– 34.

## 11 Training and Inference Time

The model was trained for a total of 57.9 hours on 16 H100 nodes, each equipped with 8 GPUs. For inference, the average time per frame is $23728 \pm 207$ ms, measured on a single A6000 GPU.

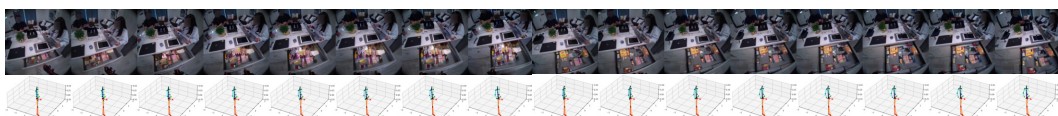

Figure 14: **Physical Realism**. When we repeatedly prompt the model to keep opening a drawer, it correctly predicts that the drawer stays in its fully opened position. This demonstrates an understanding of physical constraints.

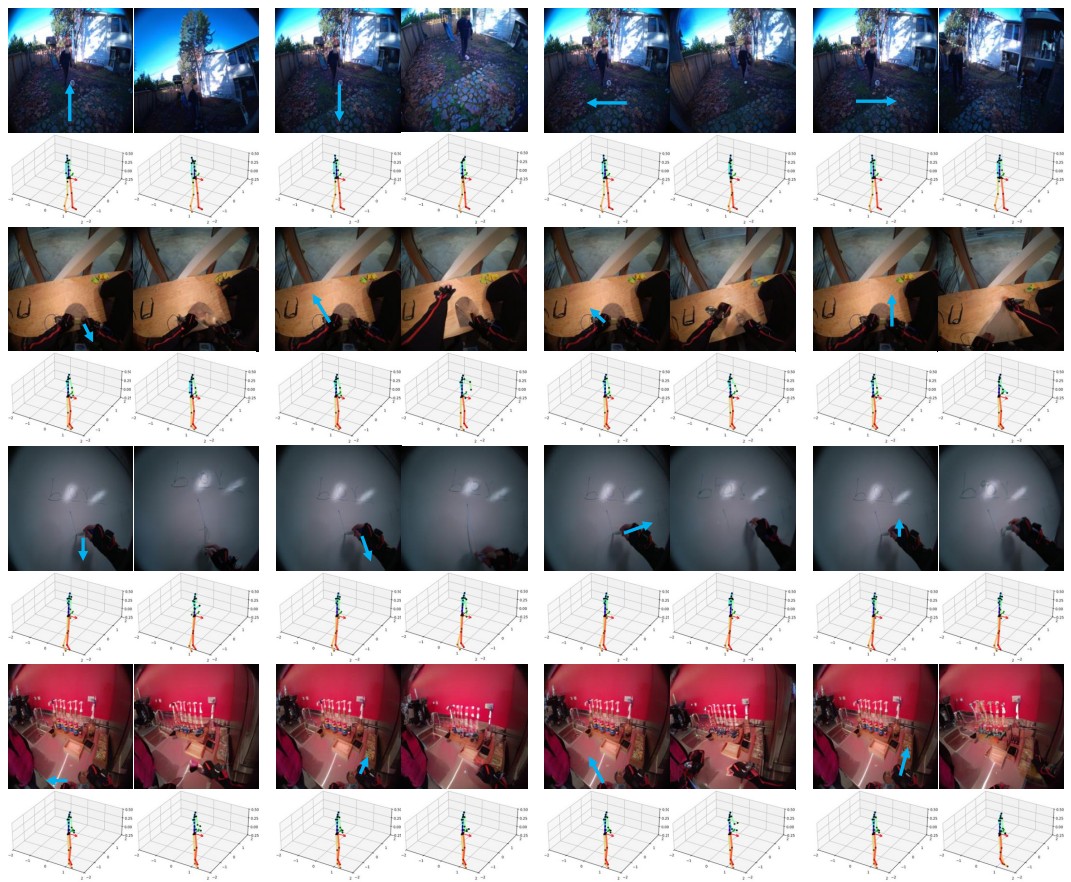

Figure 15: **Counterfactuals**. We include counterfactuals on head movement and hand movement.

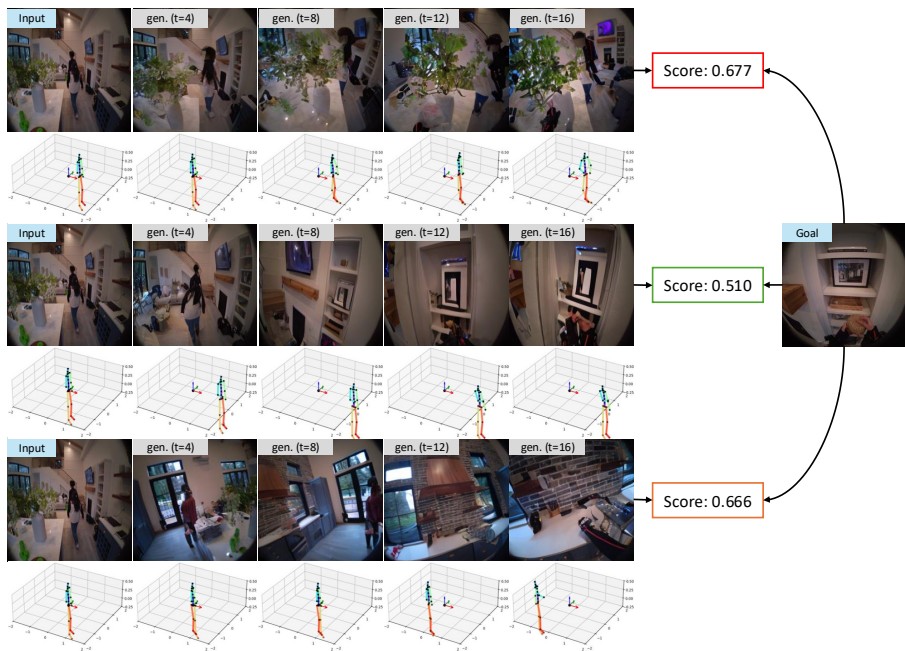

Figure 16: **Planning with Counterfactuals**. PEVA allows us to find a reasonable sequence of actions to open the refrigerator in the third row. PEVA enables us to rule out action sequences that grab the nearby plants and go to the kitchen and mix ingredients. PEVA allows us to choose the most correct action sequences that grab the box from the shelf.

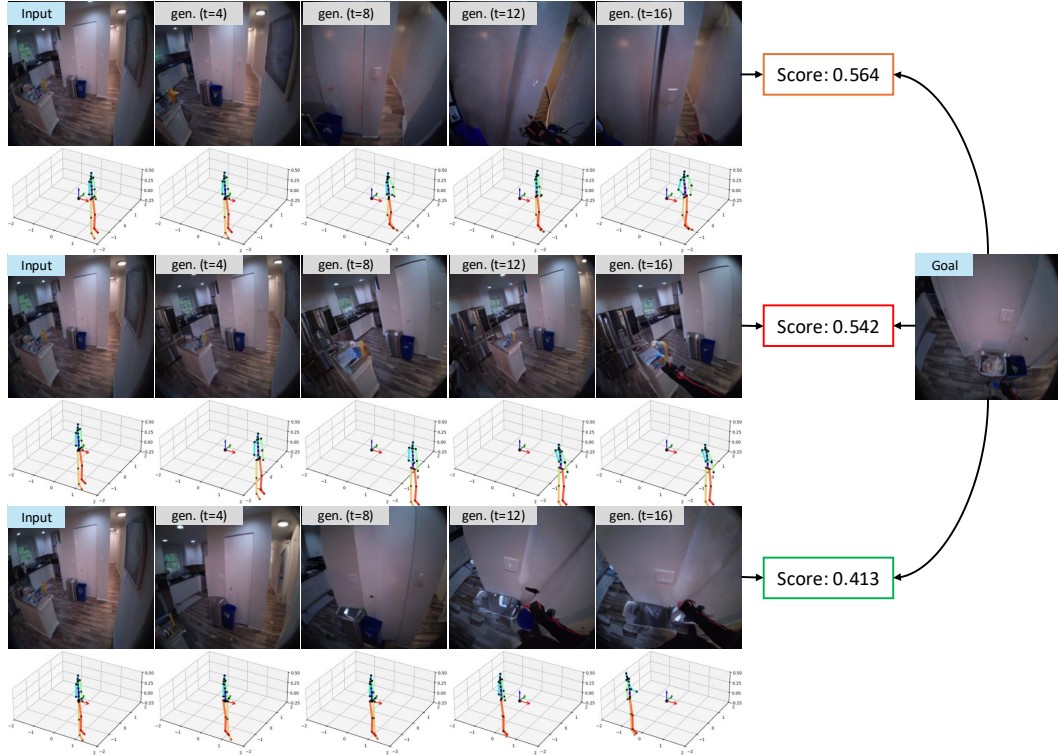

Figure 17: **Planning with Counterfactuals**. PEVA allows us to rule out action sequences that lead to the light switch in the first row and the counter in the second row. PEVA allows us to find a reasonable sequence of actions that opens the trash can and throws trash away in the third row.

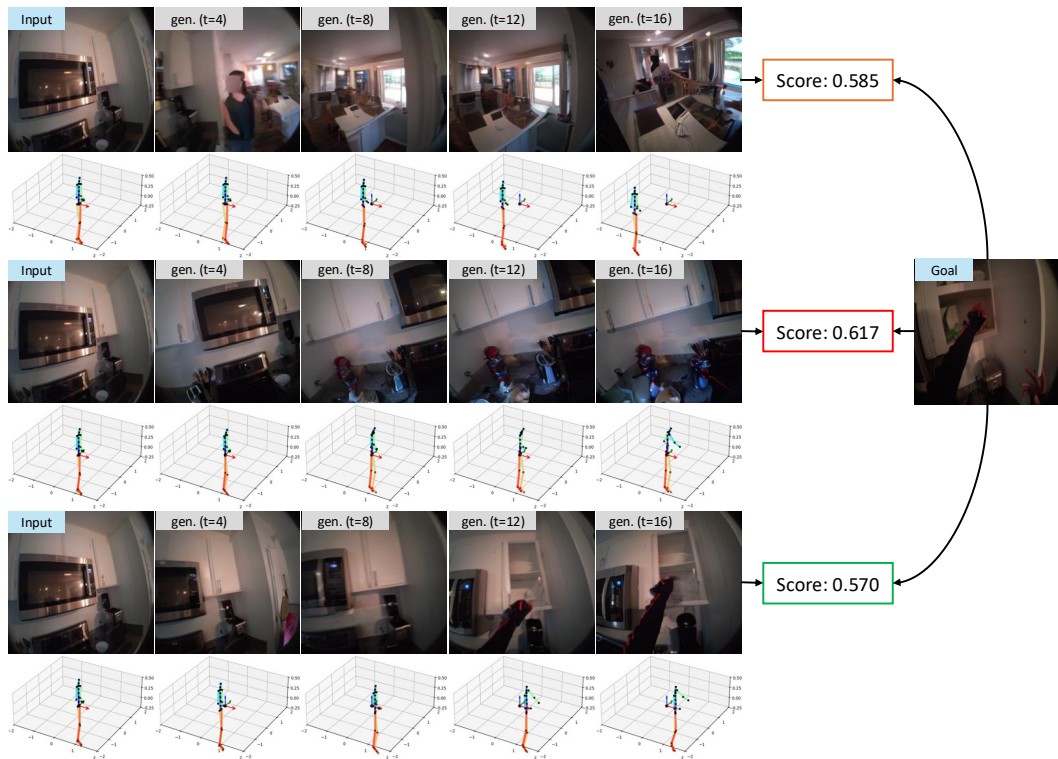

Figure 18: **Planning with Counterfactuals**. PEVA allows us to rule out action sequences that lead to the counter in the first row and the kitchen utensils in the second row. PEVA allows us to find a reasonable sequence of actions that opens the cabinet and reach in the third row.

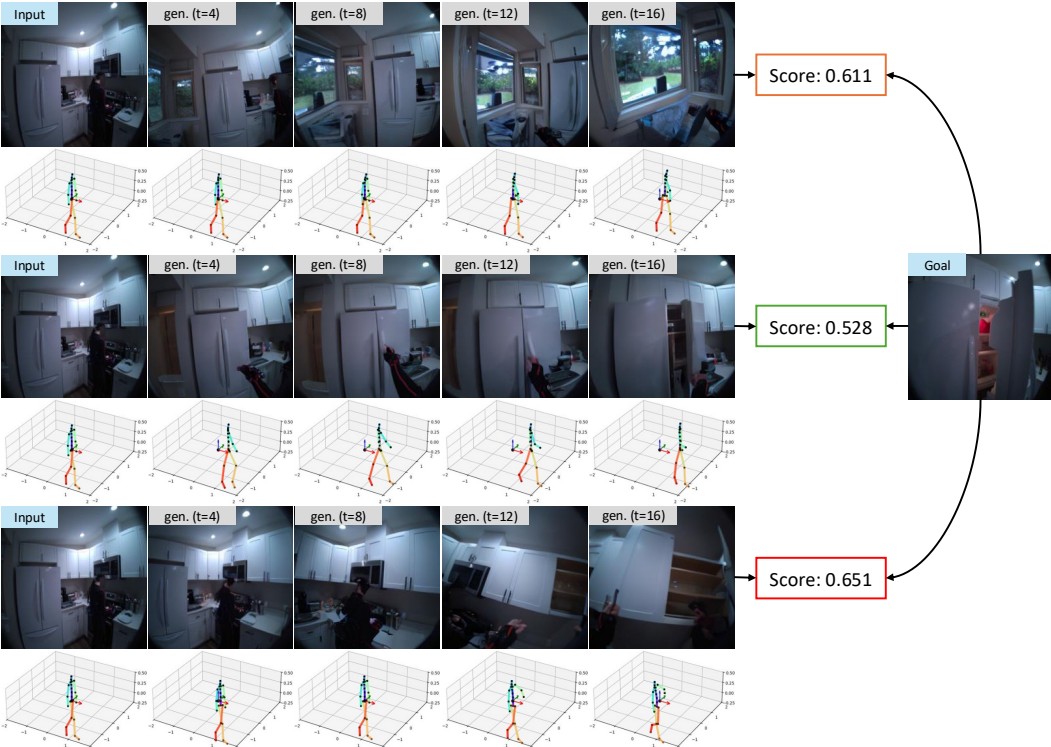

Figure 19: **Planning with Counterfactuals**. PEVA allows us to rule out action sequences that lead to the reaching into the sink in the first row and opening the cabinet in the third row. PEVA allows us to find a reasonable sequence of actions that opens the fridge in the second row.

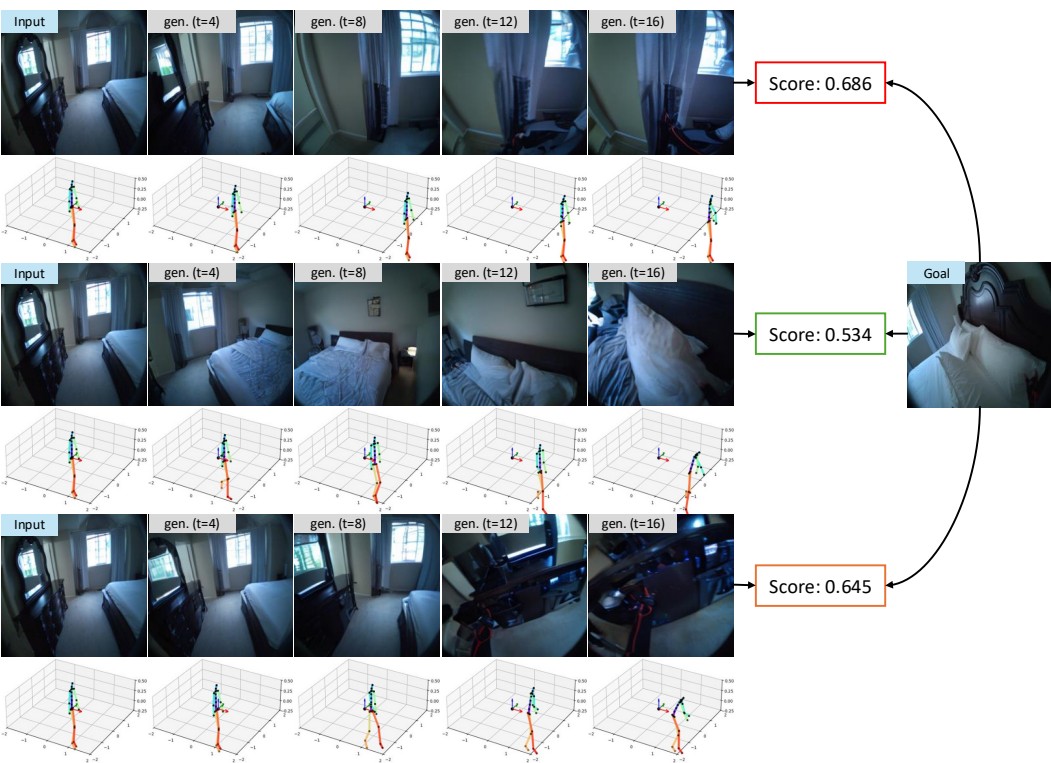

Figure 20: **Planning with Counterfactuals**. PEVA allows us to rule out action sequences that lead to the grabbing the curtains in the first row and grabbing the cabinet in the third row. PEVA allows us to find a reasonable sequence of actions that grabs the pillow in the second row.

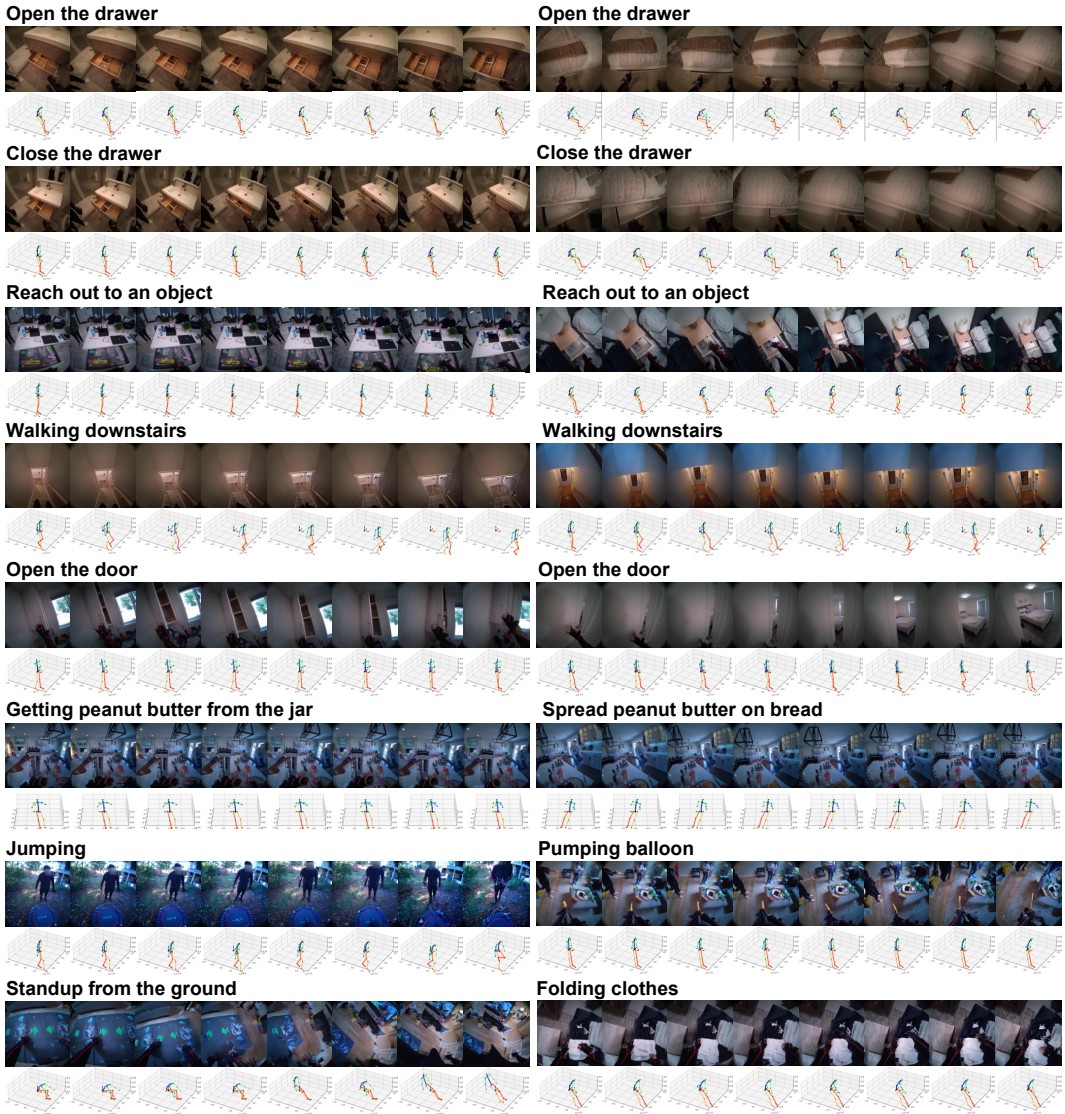

Figure 21: **Interactions**. We include video generation of interaction rich actions specified by 3D-body poses.

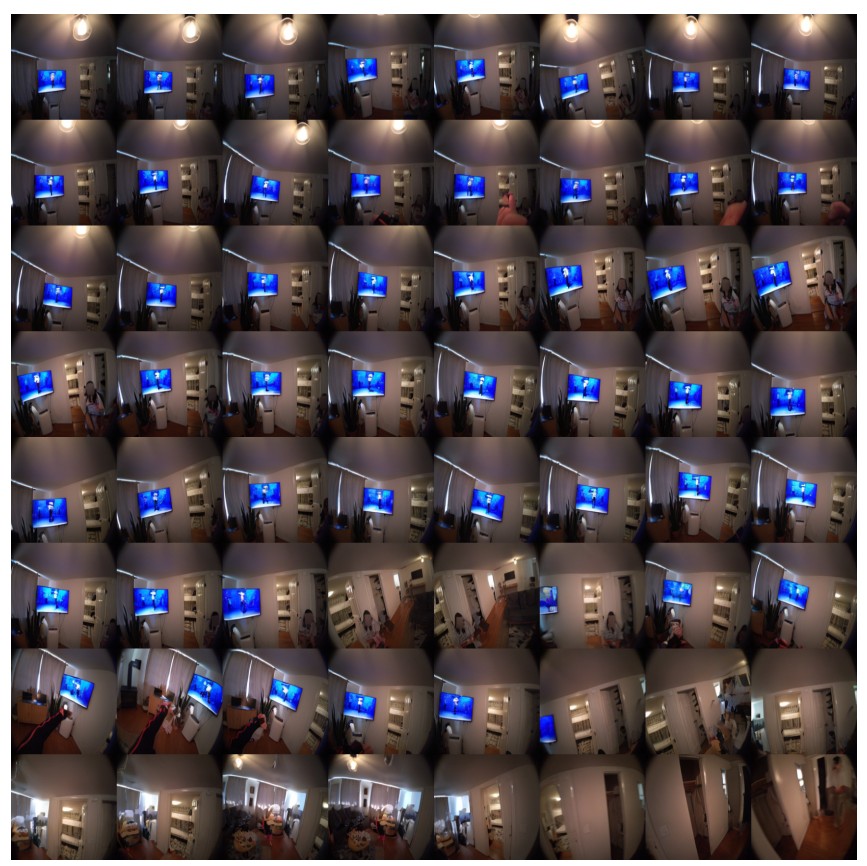

Figure 22: **Generation Over Long-Horizons**. We include 16-second video generation examples.

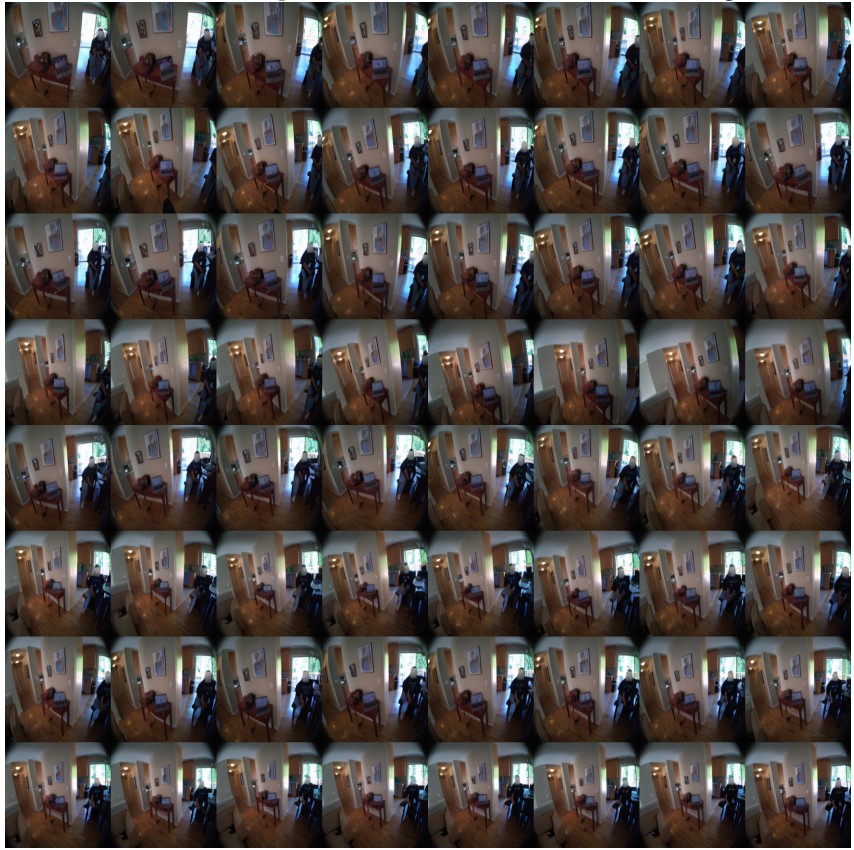

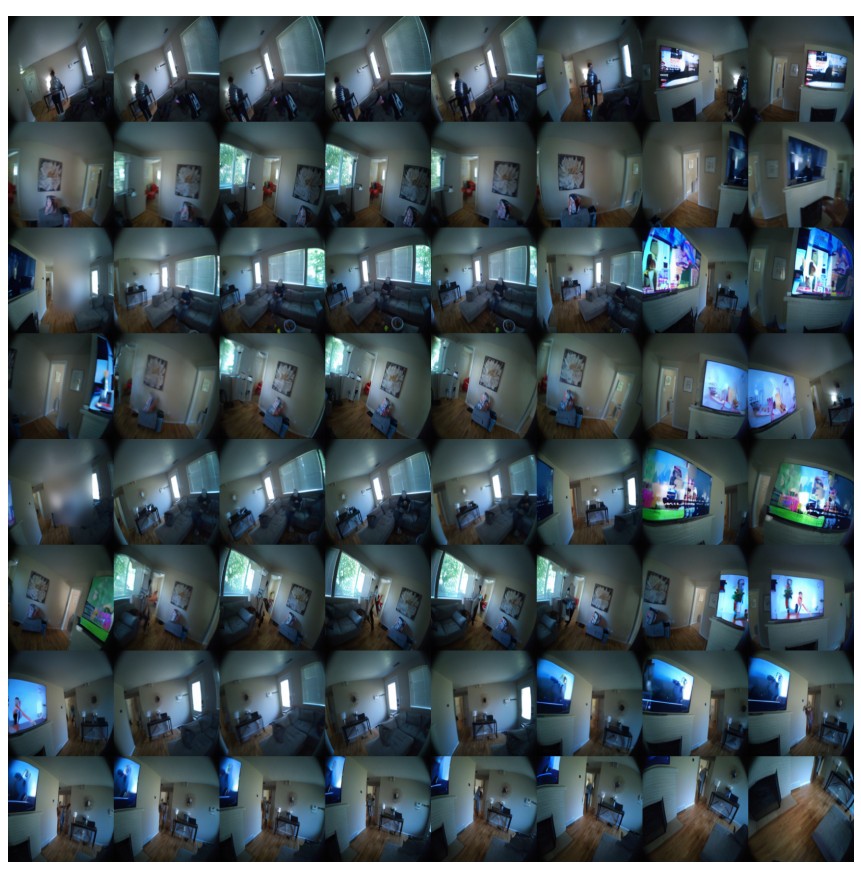

Figure 23: **Generation Over Long-Horizons**. We include 16-second video generation examples.

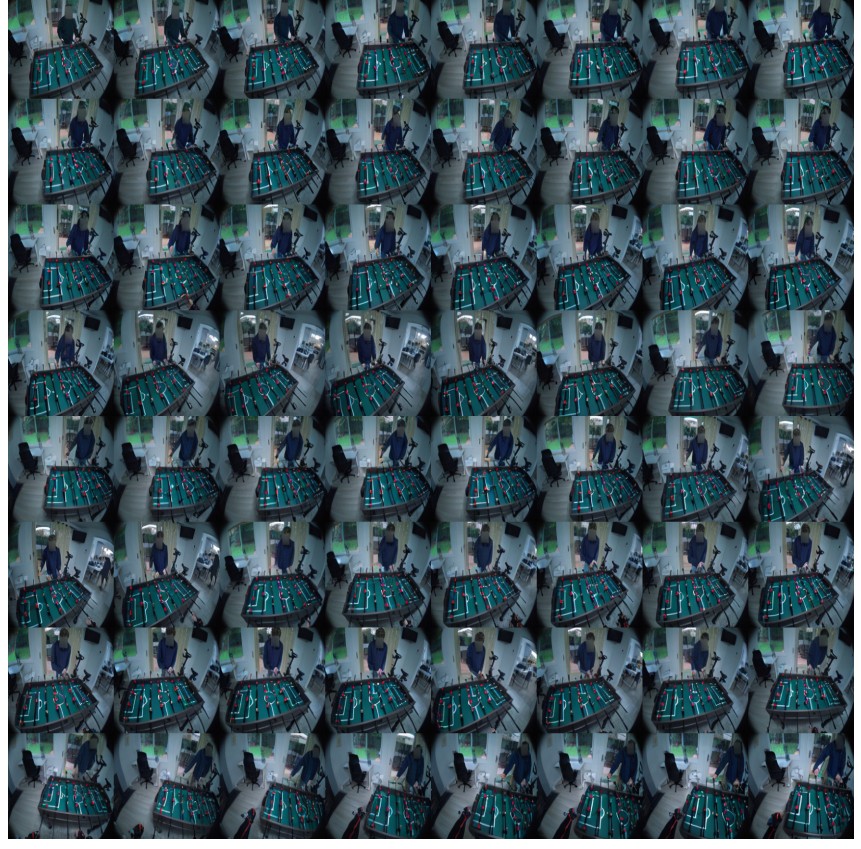

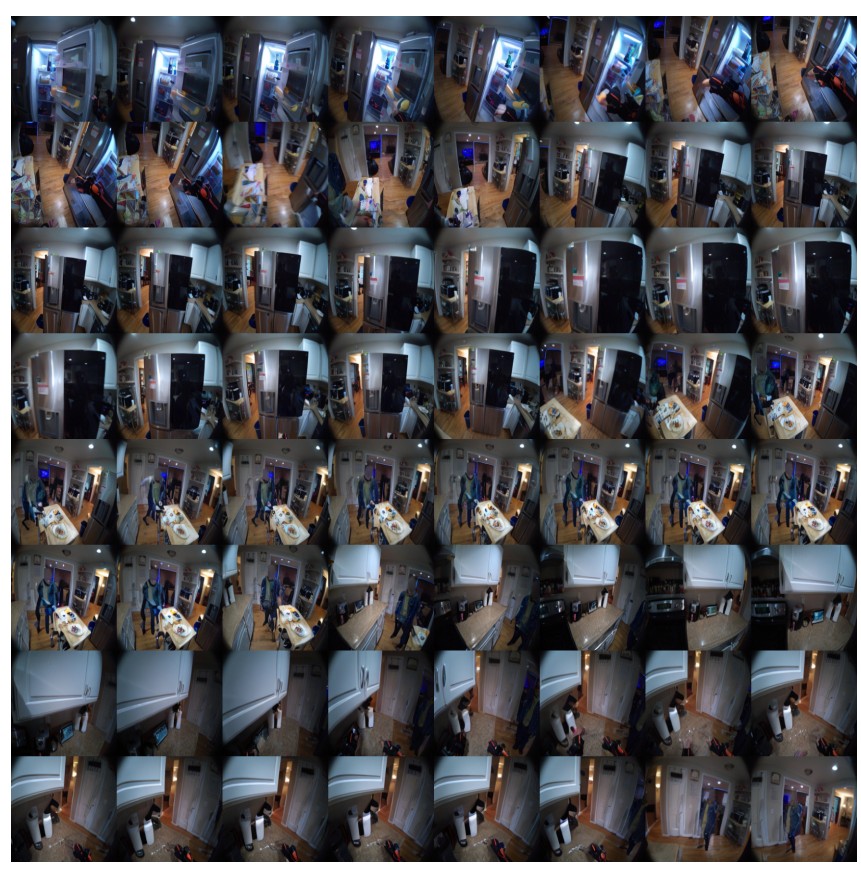

Figure 24: **Generation Over Long-Horizons**. We include 16-second video generation examples.

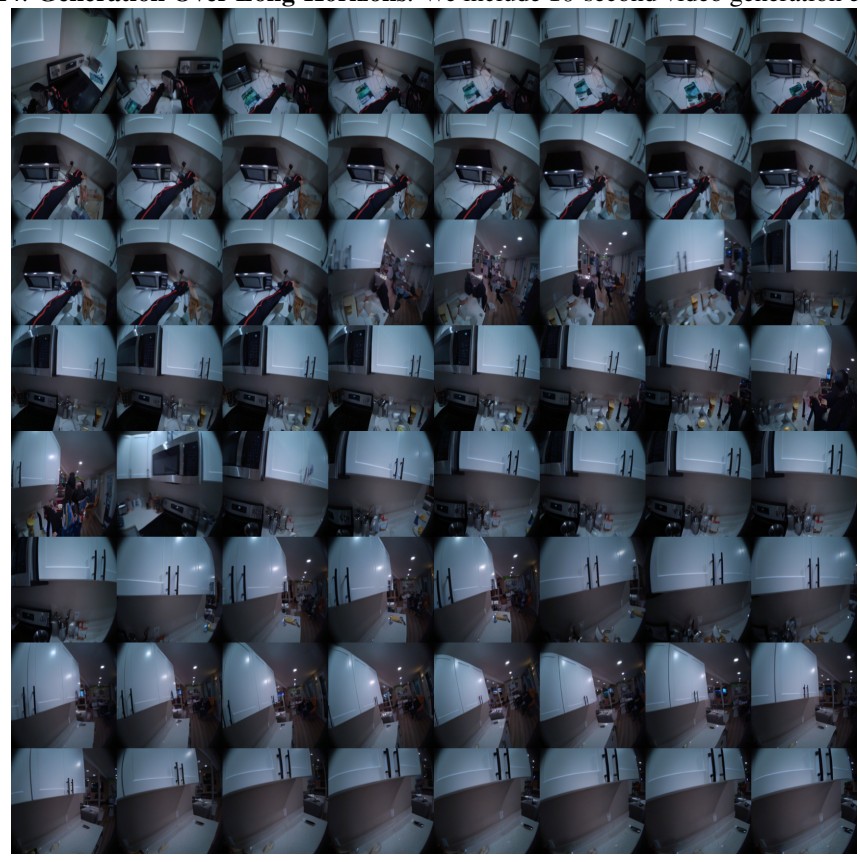

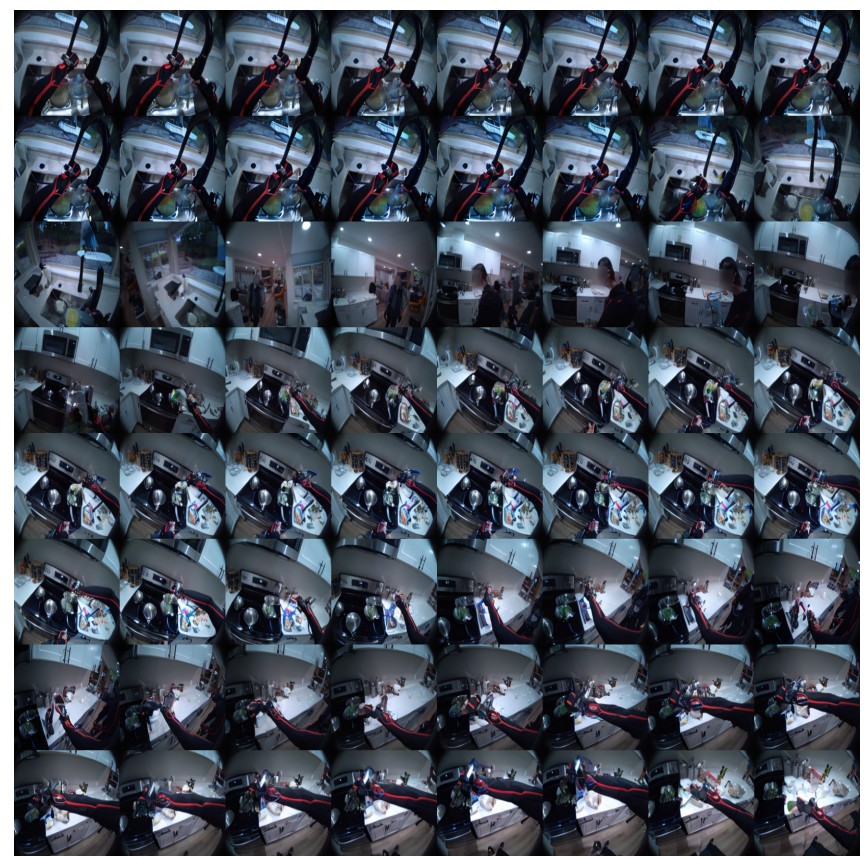

Figure 25: **Generation Over Long-Horizons**. We include 16-second video generation examples.

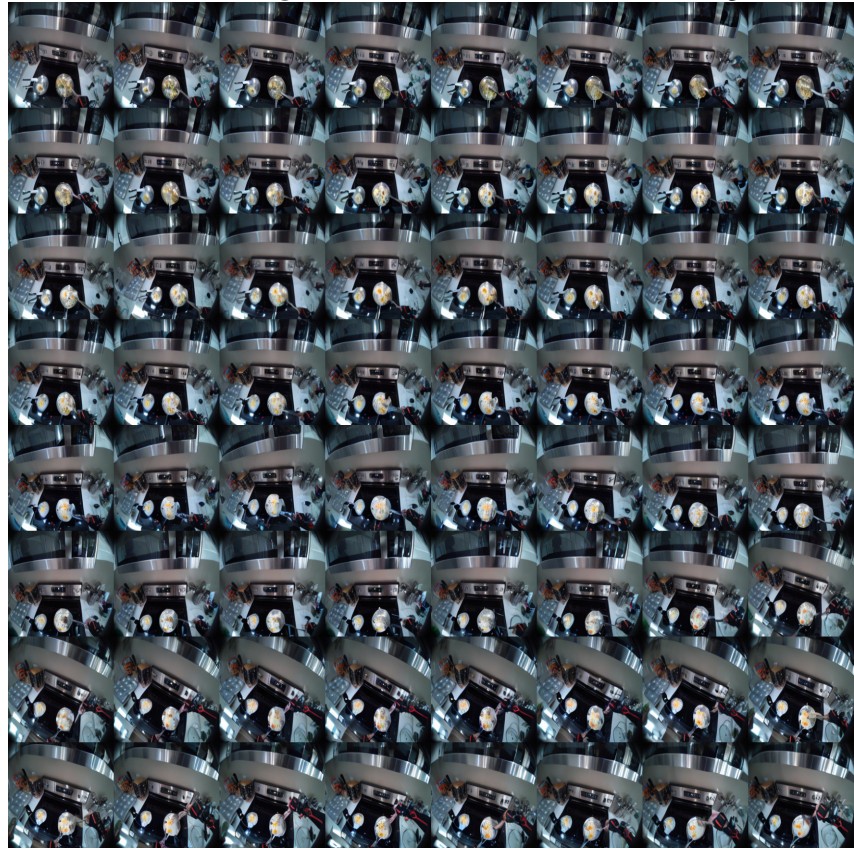

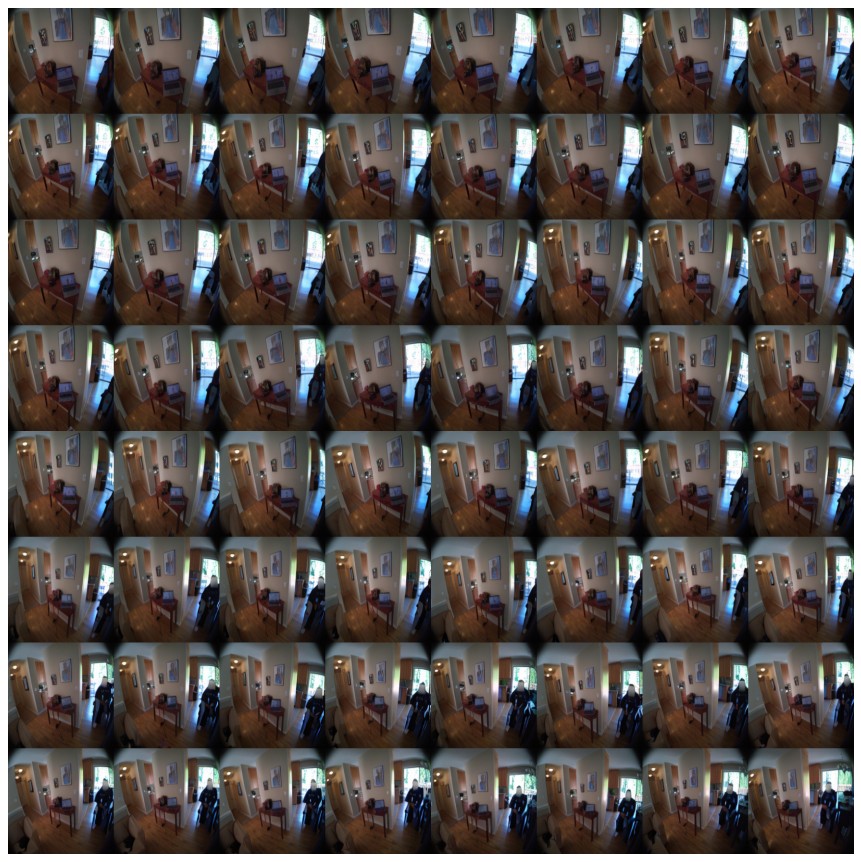

Figure 26: **Generation Over Long-Horizons**. We include 16-second video generation examples.

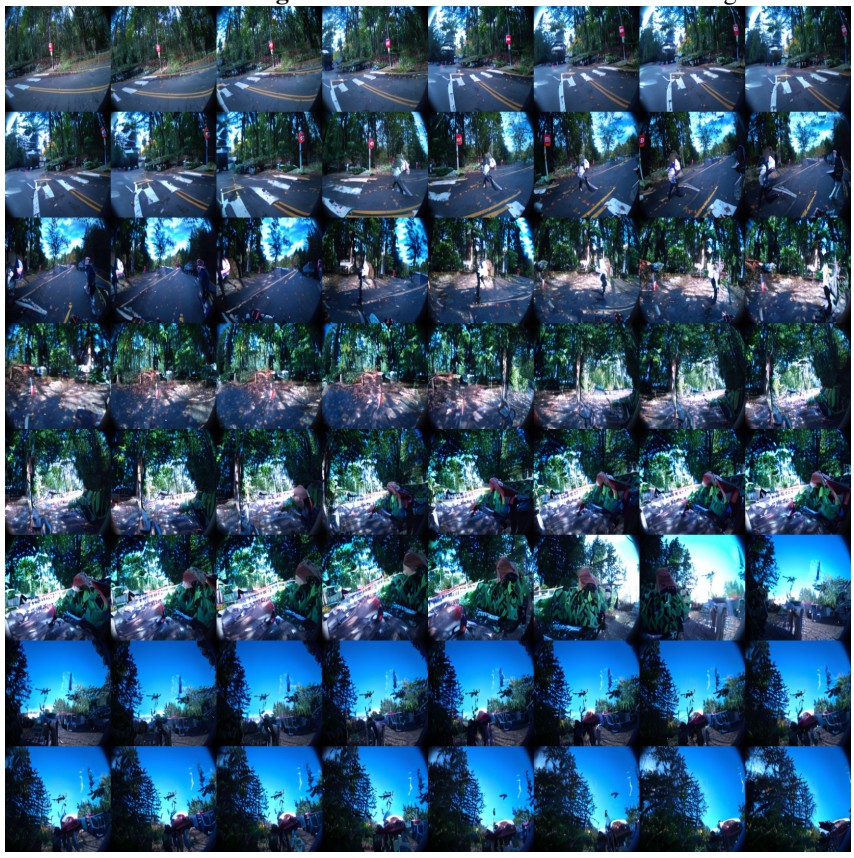

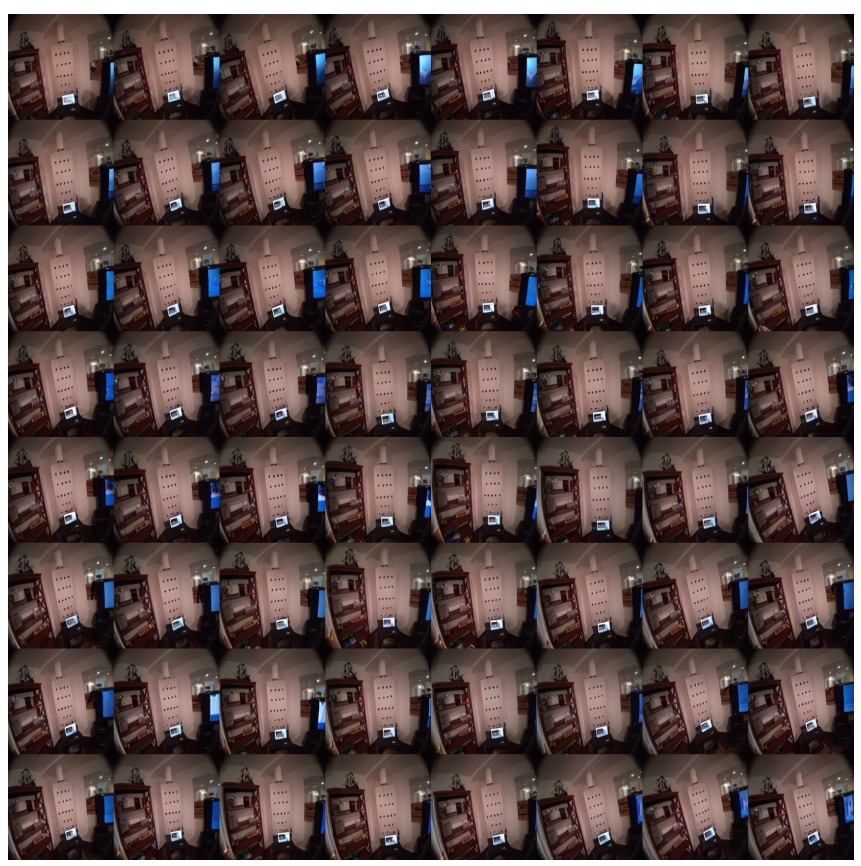

Figure 27: **Generation Over Long-Horizons**. We include 16-second video generation examples.

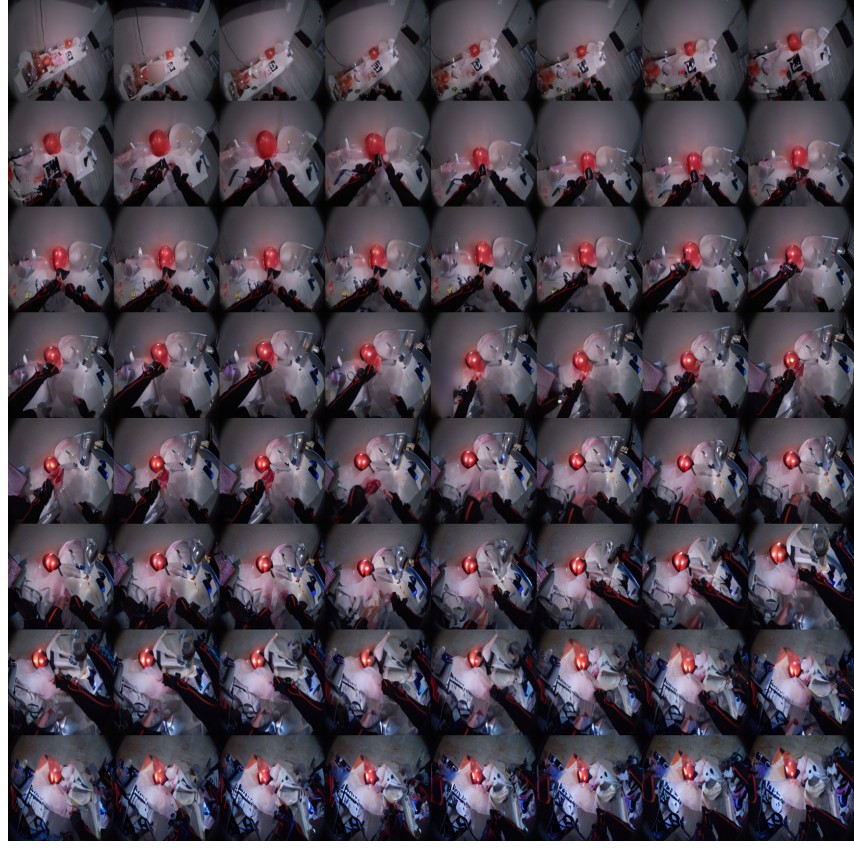

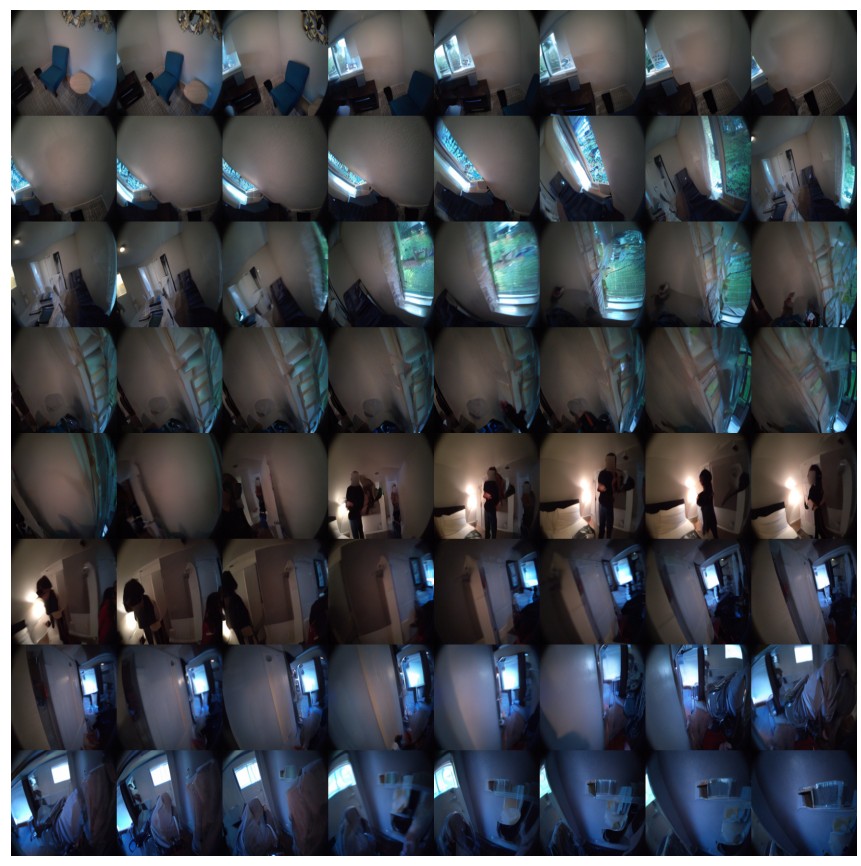

Figure 28: **Generation Over Long-Horizons**. We include 16-second video generation examples.

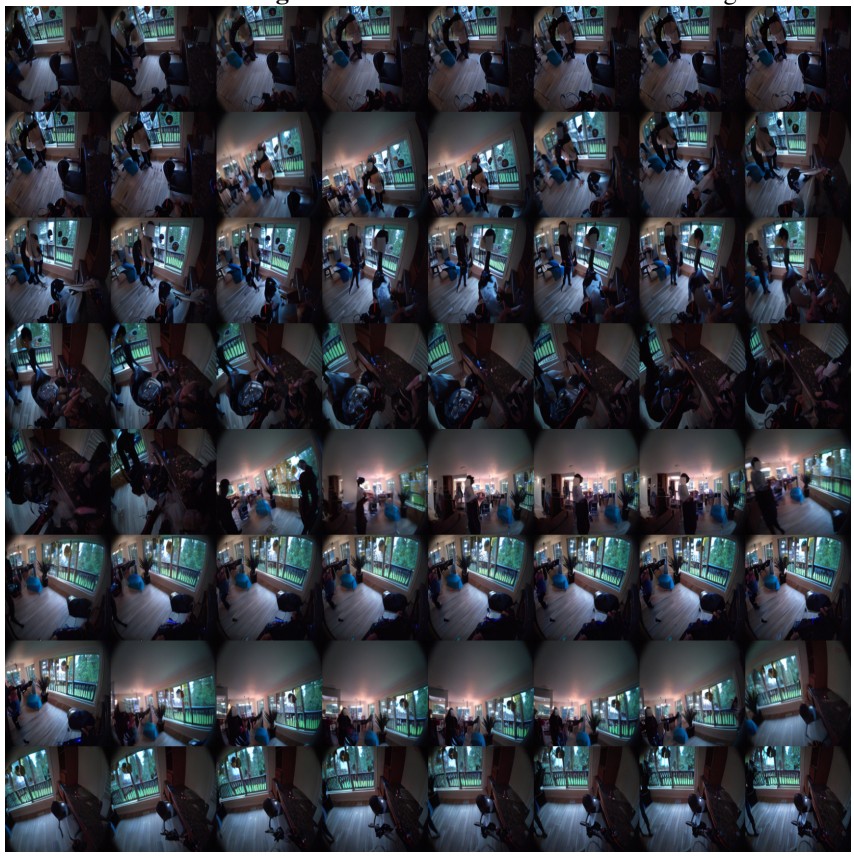

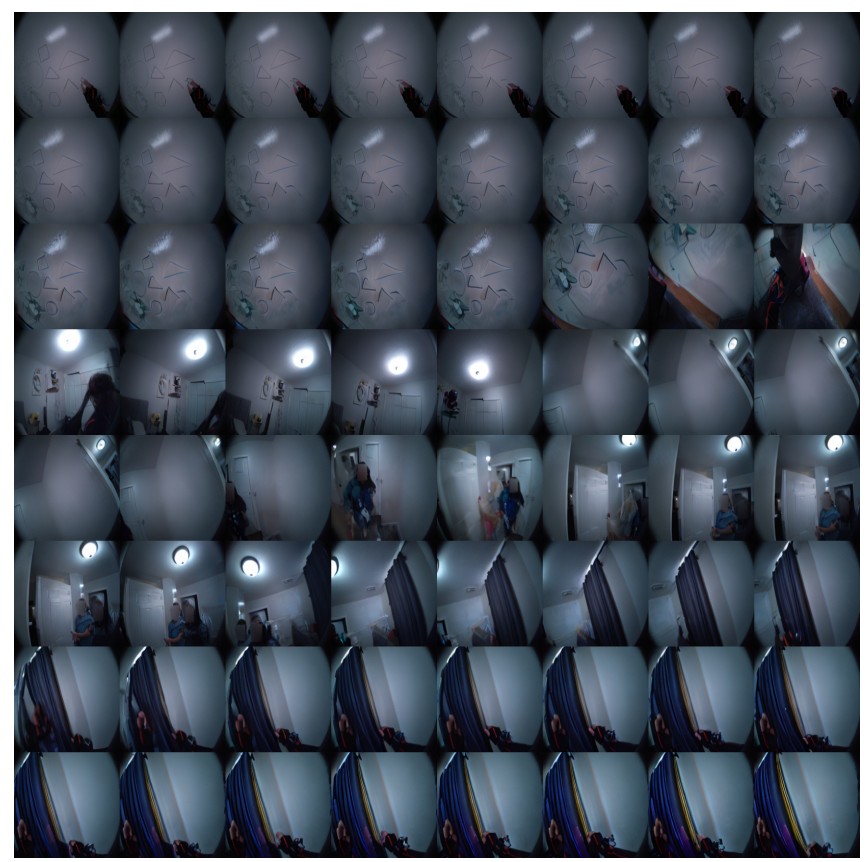

Figure 29: **Generation Over Long-Horizons**. We include 16-second video generation examples.

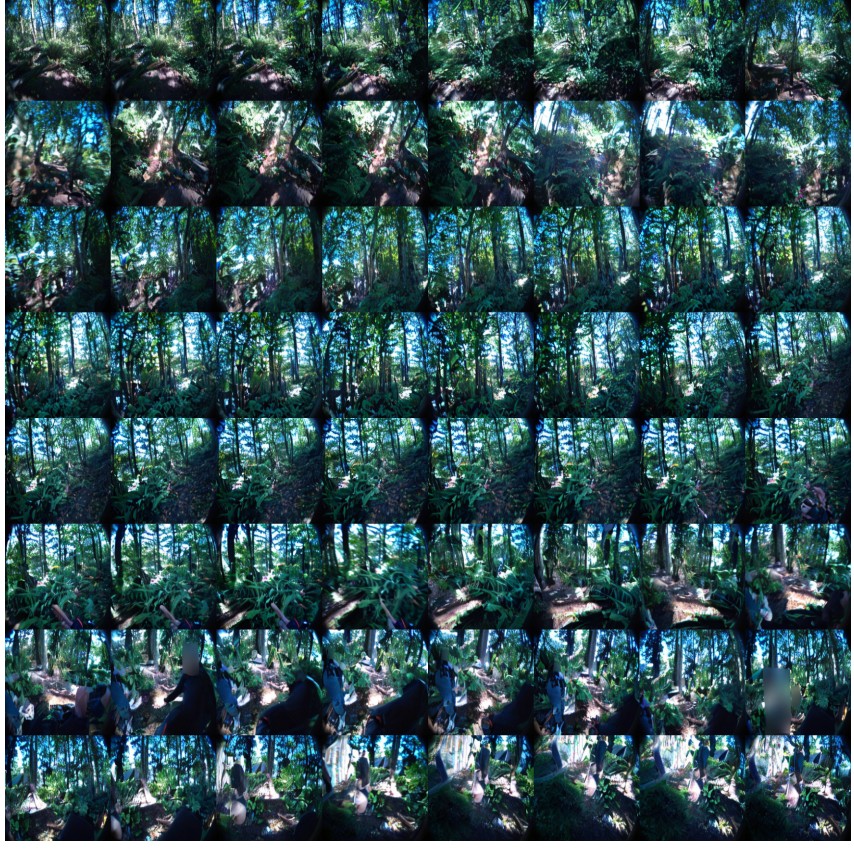

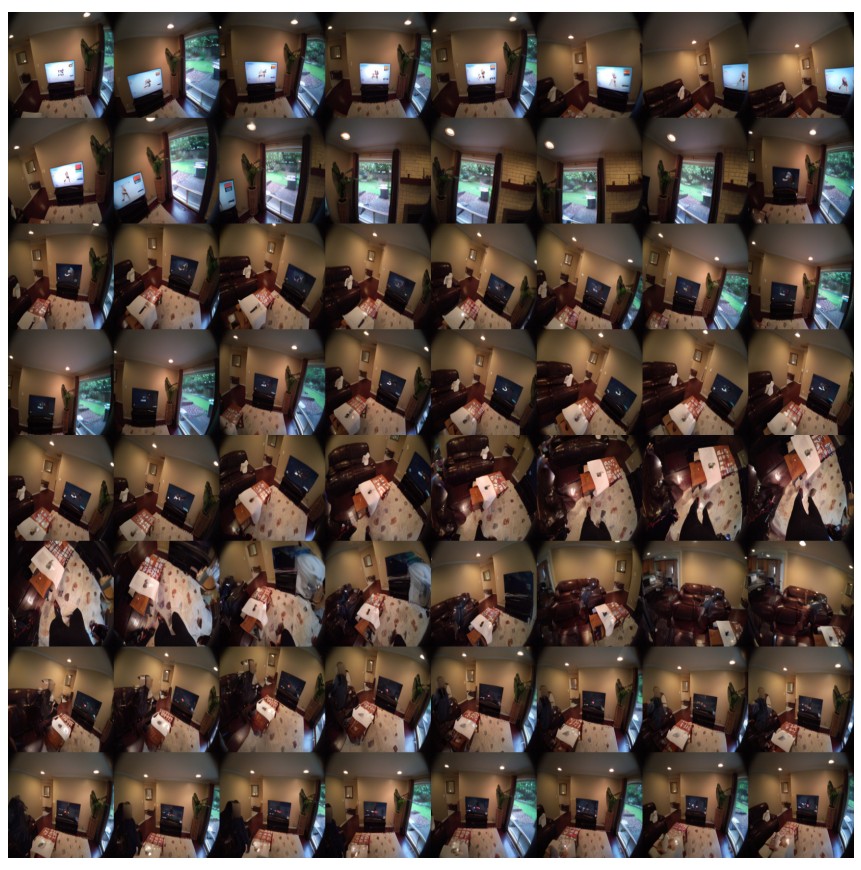

Figure 30: **Generation Over Long-Horizons**. We include 16-second video generation examples.

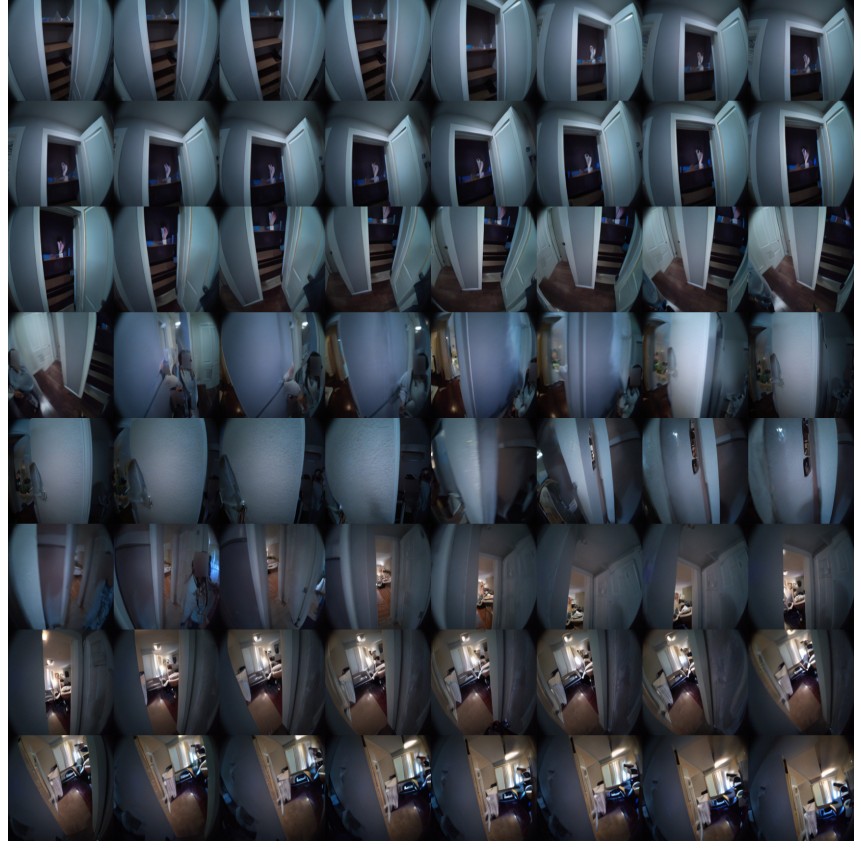

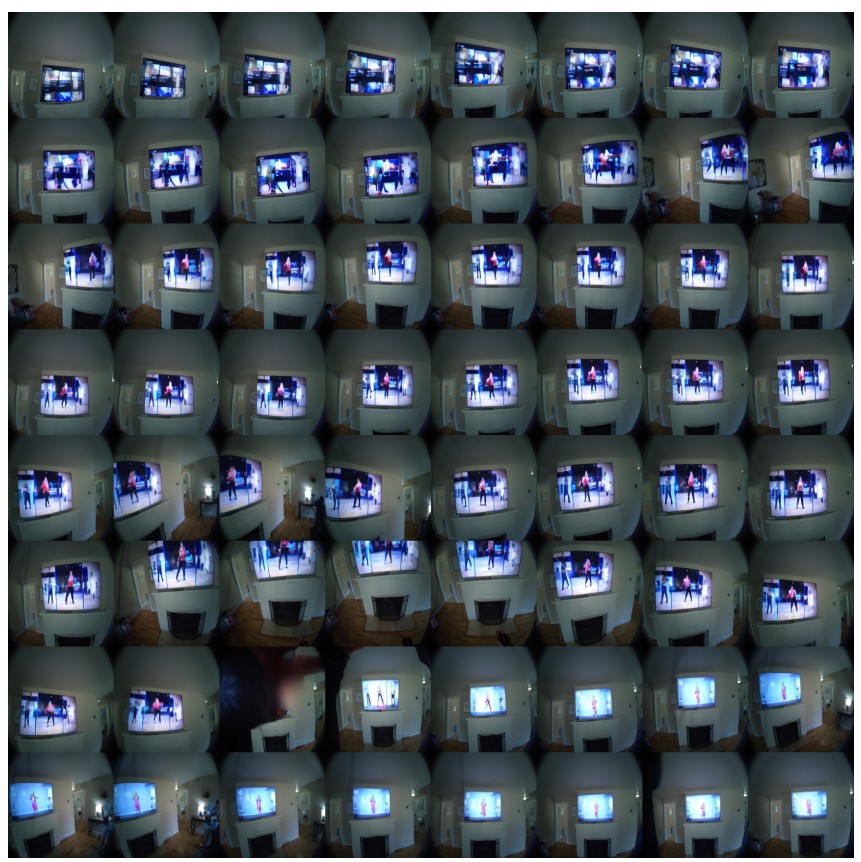

Figure 31: **Generation Over Long-Horizons**. We include 16-second video generation examples.

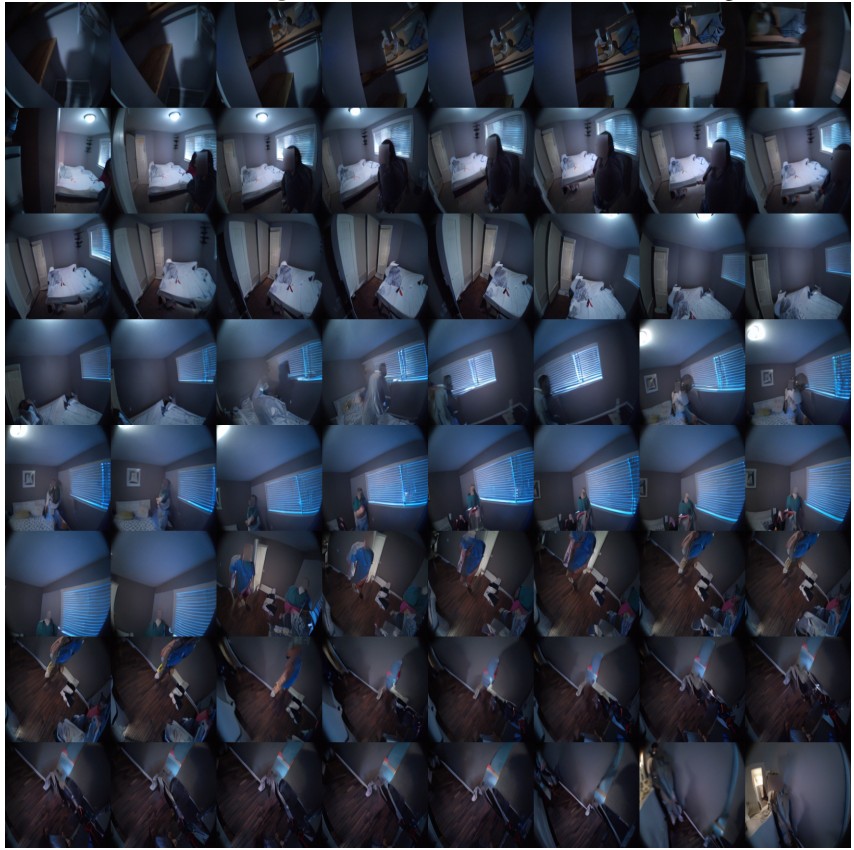

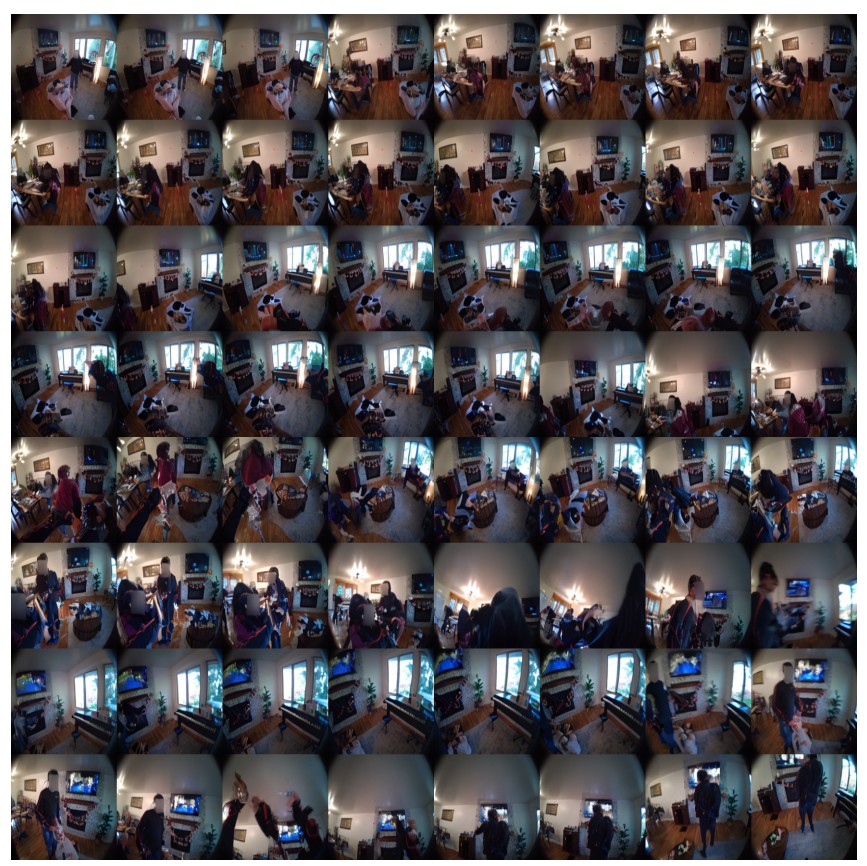

Figure 32: **Generation Over Long-Horizons**. We include 16-second video generation examples.

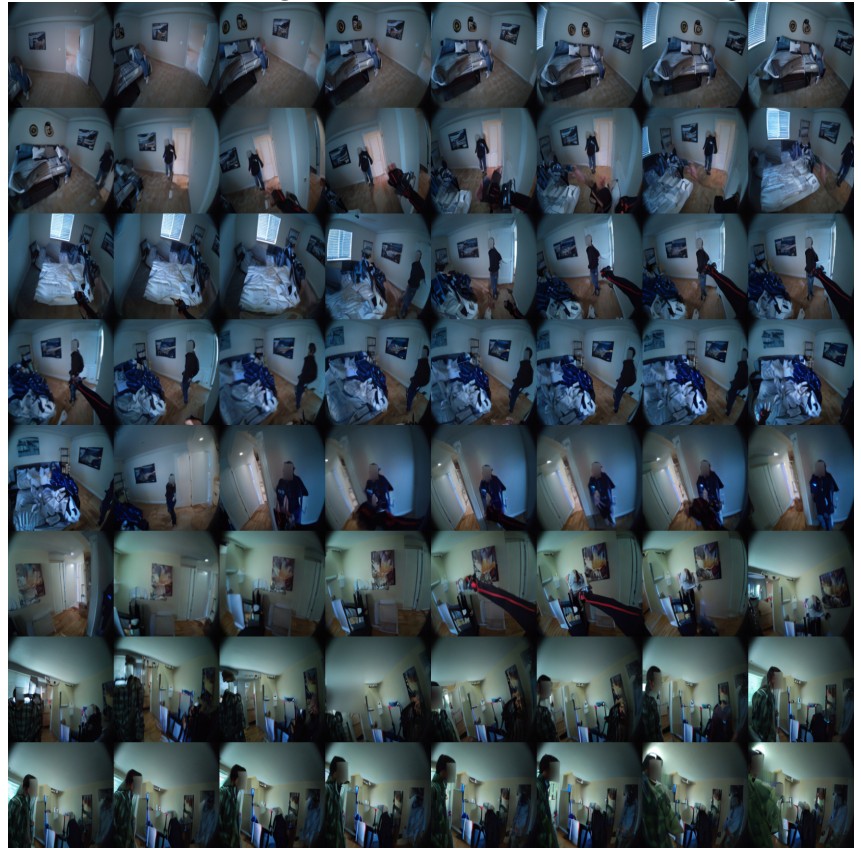

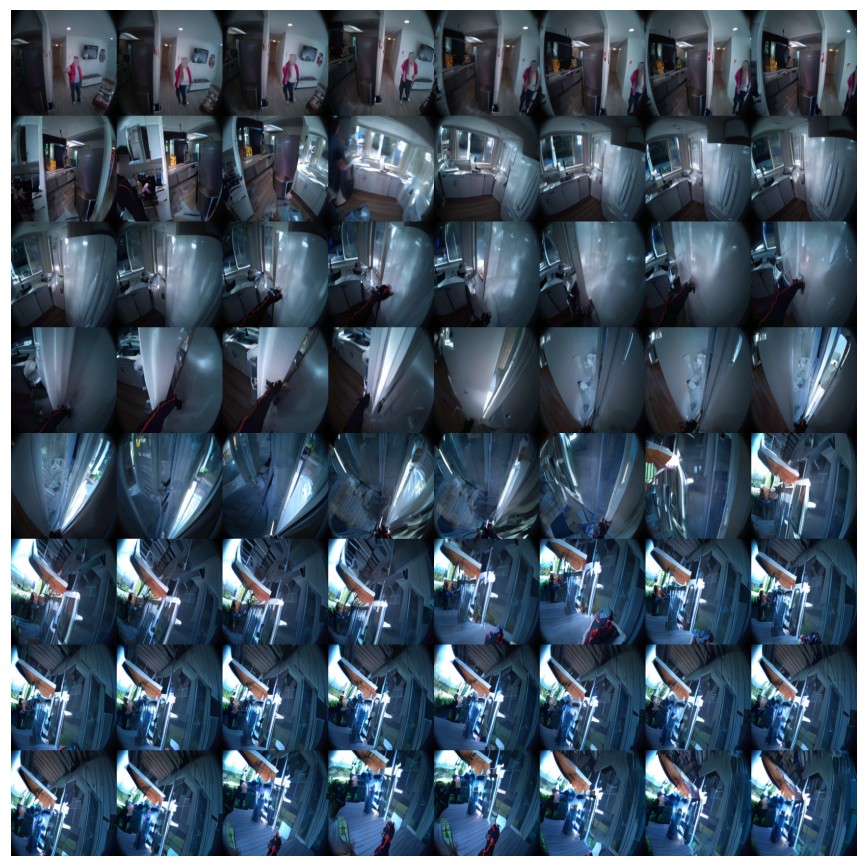

Figure 33: **Generation Over Long-Horizons**. We include 16-second video generation examples.

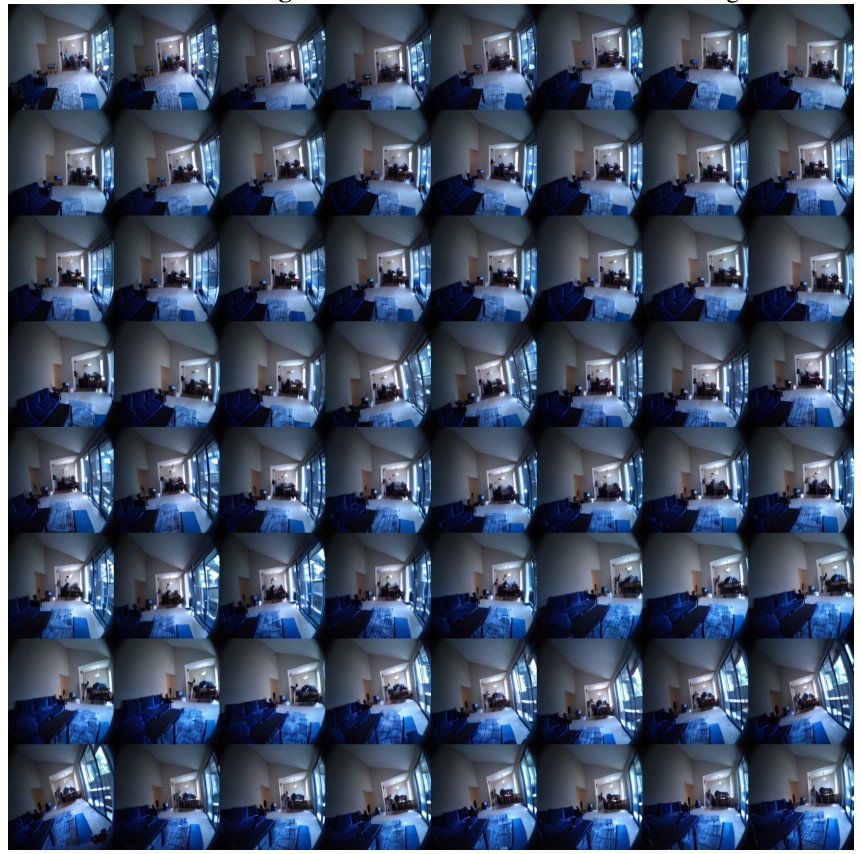

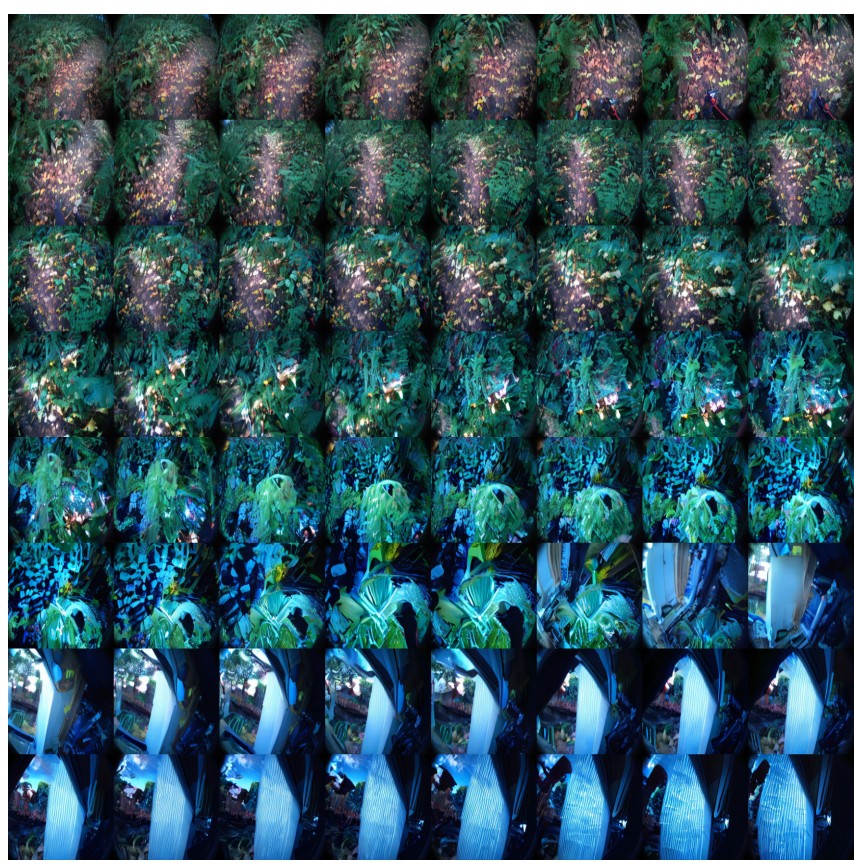

Figure 34: **Generation Over Long-Horizons**. We include 16-second video generation examples.

