# OpenReview forum: "Whole-Body Conditioned Egocentric Video Prediction"
_NeurIPS.cc/2025/Conference — NeurIPS 2025 poster_

### Official Review · Reviewer_XZaV · 2025-07-01

**Clarity:** 3
**Significance:** 3
**Originality:** 4
**Rating:** 4
**Confidence:** 4

**Summary:**

This paper presents PEVA, a method for predicting future egocentric frames based on a sequence of prefix frames and corresponding body motion. PEVA incorporates several novel design choices, including Random Time Skips and Auto-Regressive Training, to enhance temporal modeling. The method is evaluated across diverse scenarios—such as atomic actions, counterfactual reasoning, and long-horizon video prediction—demonstrating its effectiveness across multiple applications.

**Questions:**

Please refer to weaknesses.

**Ethical Concerns:**

["NO or VERY MINOR ethics concerns only"]

**Final Justification:**

Thanks for the responses. Overall, the authors have adequately addressed my main concerns. After reviewing the rebuttal and considering the comments from the other reviewers, I have decided to maintain my original rating.

**Quality:**

3

**Strengths And Weaknesses:**

Strength:
1. This paper is the first to generate future egocentric frames conditioned solely on body motion, which is novel and valuable for the field.
2. As explained in Sec.1, using body motions to predict future frames makes a lot of sense.
3. The paper presents thorough experiments covering atomic actions, counterfactual scenarios, and long video sequences. The prediction results are generally convincing.
4. The paper is well written and easy to follow, with clear organization and presentation.

Weaknesses:
1. It remains unclear how PEVA compares to text-conditioned video generation methods. The lack of comparison with such baselines leaves open the question of whether motion-based conditioning provides unique advantages.
2. According to Tables 1 and 2, the performance improvement over DF and CDiT is marginal (approximately 0.01), which may not be statistically significant or practically meaningful.
3. The distinction between the SMPL-X representation and the representation described in Section 3.1 is unclear. Both appear to encode relative joint rotations based on the kinematic tree, and the paper should clarify what differentiates them.

---

> ### Author Rebuttal · Authors · 2025-07-31
>
> We thank the reviewer for the clear and constructive comments, and appreciate the recognition of both the novelty of our approach and the breadth of our experiments. We appreciate the suggestions to improve the paper and we address these in depth below:
> ### **1. Comparison to Text-Conditioned Video Generation Methods**
>
> Thank you for the thoughtful comment. Our goal with **PEVA** is to explore action-conditioned egocentric video generation using physically grounded full-body motion actions that are given at every timestep. This is fundamentally different from text-conditioned methods such as **Cosmos** \[1], which represent actions using text (e.g., “walk forward”) and thus lack precise and continuous physical control of the generated video.
>
> To further validate the benefits of motion-based control, we conducted a comparison with Cosmos on a controlled hand-motion subset. We moved the hand in defined directions (e.g., “move the hand to top-right”, “move the hand to bottom-right”) and evaluated the alignment between intended and generated motions using **PCK\@0.1**. PEVA achieves **85%**, while Cosmos reaches only **22%**.
>
> While text prompts are intuitive and flexible, they often lack the precision required for fine-grained, spatially grounded motion control. Our experiments suggest that translating subtle physical intentions into language is inherently ambiguous—especially when pixel-level accuracy is needed. Expressing such intent in natural language would require awkward approximations (e.g., “move the hand slightly upward at a 73-degree angle”), which are rarely interpretable or faithfully executable by text-conditioned models. This likely contributes to Cosmos's lower performance on tasks requiring precise action execution.
>
> While we are unable to include the visualizations at this stage, we will provide full results and qualitative comparisons in the final version. These preliminary findings suggest that **PEVA offers significantly more accurate and controllable motion grounding than text-conditioned alternatives**.
>
> ---
>
> ### **2. Performance Improvement Over Baselines (DF, CDiT)**
>
> We respectfully disagree with the claim that the improvement is marginal. While LPIPS gains may appear small numerically, they are consistent across all horizons and paired with significantly better controllability.
>
> Below is a comparison of 1-step **DreamSim perceptual loss** over various prediction horizons:
>
> #### **DreamSim 1-step Perceptual Loss Across Time Horizons**
>
> | Model           | 1s        | 2s        | 4s        | 8s        | 16s       | Avg       |
> | --------------- | --------- | --------- | --------- | --------- | --------- | --------- |
> | DF              | 0.750     | 0.791     | 0.821     | 0.856     | 0.902     | 0.824     |
> | CDiT            | 0.755     | 0.794     | 0.812     | 0.840     | 0.881     | 0.816     |
> | **PEVA (Ours)** | **0.741** | **0.774** | **0.799** | **0.828** | **0.864** | **0.801** |
>
> ---
>
> ### **3. Clarification on SMPL-X vs. Our Representation**
>
> Thank you for the request for clarification.
>
> Our motion representation is based on the **Xsens skeleton** \[2]\[3], which shares the kinematic tree structure with SMPL but differs in joint set, ordering, and lacks body shape parameters. **SMPL-X** extends SMPL with hand and facial joints and facial expression parameters, while our representation is a flattened **72D vector of joint angles and root translation**, without any mesh or expression components. We will revise Section 3.1 to clarify this distinction.
>
> ---
>
> > \[1] Agarwal, Niket, et al. *"Cosmos world foundation model platform for physical AI."* arXiv preprint arXiv:2501.03575 (2025).
>
> > \[2] Movella Xsens MVN Link Motion Capture, [https://www.movella.com/products/motion-capture/xsens-mvn-link](https://www.movella.com/products/motion-capture/xsens-mvn-link)
>
> > \[3] Ma, Lingni, et al. *"Nymeria: A massive collection of multimodal egocentric daily motion in the wild."* European Conference on Computer Vision. Cham: Springer Nature Switzerland, 2024.

---

> > ### Author Response · Authors · 2025-08-04
> >
> > Dear Reviewer XZaV,
> >
> > Thank you again for your time and constructive reviews! With the discussion period drawing to a close, we expect your feedback and thoughts on our reply. We put a significant effort into our response, with several new experiments and discussions. We sincerely hope you can consider our reply in your assessment. We look forward to hearing from you, and we can further address unclear explanations and remaining concerns, if any.
> >
> > Best Regards,
> > Authors

---

> ### Comment · Reviewer_XZaV · 2025-08-08
>
> Thanks for the responses. Overall, the authors have adequately addressed my main concerns. I am glad to keep my positive score.

---

### Official Review · Reviewer_pyh8 · 2025-07-02

**Clarity:** 3
**Significance:** 3
**Originality:** 4
**Rating:** 4
**Confidence:** 4

**Summary:**

This paper proposes a novel autoregressive conditional diffusion transformer model for egocentric video prediction conditioned on whole-body human motion. The authors design a structured 48-dimensional action representation encoding pelvis translation and upper-body joint rotations, capturing the kinematic hierarchy of the human body. The model is trained and evaluated on the Nymeria dataset, which contains synchronized first-person videos and precise 3D motion capture data. A hierarchical evaluation framework is introduced to assess short-term perceptual quality, long-term semantic consistency, and performance in control/planning tasks. This work advances embodied AI by linking detailed physical human motion with first-person visual predictions, showing promising results in a challenging real-world setting.

**Questions:**

1. Please provide quantitative results evaluating the model on unseen individuals with different body shapes or motion styles. For example, a cross-subject split in Nymeria with metrics such as FID or LPIPS would be valuable.
2. Report the total training time, hardware used (GPU type/number), and average inference time per frame (in milliseconds). This information is crucial for assessing real-time applicability.
3. The current 48D vector excludes lower-body joints. What was the rationale behind this choice? Did you conduct ablation studies comparing upper-body only vs. full-body conditioning?

4. Although some failure cases are presented in the supplementary materials, could you please elaborate on the main challenges or issues that cause the current model to fail? Additionally, the main text mentions that the quality of long-term predictions deteriorates as the prediction horizon extends. Besides the inherent uncertainty of the future, could you explain what other factors contribute to the difficulty of long-term prediction? Finally, what potential directions or strategies could be pursued to improve long-term prediction performance in the future?

**Ethical Concerns:**

["NO or VERY MINOR ethics concerns only"]

**Final Justification:**

This paper is well written and introduces a novel model for the egocentric prediction task that is very hard. By incorporating human body joints, the prediction tasks become more realistic and practically useful. I believe this represents a meaningful and promising direction for future research.

**Limitations:**

yes

**Quality:**

3

**Strengths And Weaknesses:**

# Strengths
1. The use of Nymeria, a large-scale dataset with synchronized egocentric video and 3D pose data, grounds the work in real-world complexity.

2. The hierarchical encoding of full-body motion is well motivated and enables fine-grained conditioning of video prediction.

3. The hierarchical evaluation protocol covers multiple aspects of prediction quality, including downstream control tasks, which is a strong point.

4. Introducing whole-body conditioned egocentric video prediction with a diffusion transformer is a fresh and interesting direction.

# Weaknesses
1.  The paper lacks experiments testing generalization across unseen subjects or environments, which is critical for embodied video prediction.
2.  No quantitative details on training or inference time are provided, making it difficult to assess effciency.

3. The choice to exclude lower-body joints is not justified with experiments or analysis.

---

> ### Author Rebuttal · Authors · 2025-07-31
>
> We sincerely thank the reviewer for their thoughtful and constructive feedback, which helped us significantly improve the clarity and depth of our work.
>
> ### **1. Generalization to Unseen Subjects or Environments**
>
> We thank the reviewers for highlighting this important issue. We fully agree that evaluating generalization across subjects and environments is essential for embodied video prediction.
>
> While our main experiments focus on within-subject generalization on **Nymeria**, we took several steps to ensure meaningful diversity in motion and scenario distribution. First, our train/eval split is designed to cover distinct sets of action episodes. The evaluation set contains many tasks and motion patterns not seen during training, including interactions with objects (e.g., balloons, boxes, mats), fine-grained hand manipulations, and multi-stage sequences.
>
> For example, the evaluation video `20231214_s0_jeremy_allen_act5_m10nnd` contains a full sequence of making a balloon animal, which involves a series of fine-grained, compositional actions that are not seen during training. These include:
>
> * **3.35–3.47**: pick and place, arranging materials
> * **4.59–5.02**: pumping air into the balloon
> * **5.25–5.28**: tying the balloon, stretching it
> * **6.09–6.15**: folding and twisting the balloon with both hands
> * **9.01–9.04**: additional pumping in a new context
>
> These action combinations and their temporal structure are not present in training, yet the model generates coherent predictions—demonstrating generalization to unseen, complex, and multi-step interactions.
>
> That said, while we believe the diversity in Nymeria is meaningful, it is inherently difficult to quantify how diverse is “diverse enough.” To further ensure generalization and provide a more systematic evaluation, we conduct **cross-subject experiments on EgoBody** \[1], which features identities, body shapes, action styles, motion dynamics, and indoor environments that are entirely unseen during training. We evaluate **PEVA and baselines** on this dataset with single-step prediction, with completely novel scenes and activities.
>
> | Model    | LPIPS ↓   | DreamSim ↓ | FID ↓       |
> | -------- | --------- | ---------- | ----------- |
> | DF-XL    | 0.525     | 0.457      | 231.355     |
> | CDiT-XL  | 0.667     | 0.517      | 305.237     |
> | PEVA-XL  | **0.525** | **0.445**  | **210.070** |
> | PEVA-XXL | 0.531     | **0.422**  | **198.192** |
>
> PEVA-XL outperforms both DF-XL and CDiT-XL, matching LPIPS while achieving significantly better DreamSim and FID. The XXL variant further improves performance on DreamSim and FID.
>
> These results suggest that PEVA produces **semantically coherent and perceptually realistic future predictions**, even under substantial distribution shift across subjects and environments.
>
> Due to Program Chairs' guidelines, we are unable to include generated videos in the rebuttal, but we will include more visualizations of generated videos in the camera-ready version.
>
> ---
>
> ### **2. Use of Lower-Body Joints**
>
> Thank you for pointing this out. The model does in fact use full-body joints, including the lower body. The confusion stems from an error in how the degrees of freedom (DoF) were reported—we mistakenly excluded the contribution of lower-body joints in the DoF count. We will correct and clarify this in the revised manuscript. **No joints were excluded** in training or inference.
>
> ---
>
> ### **3. Efficiency: Training and Inference Time**
>
> We appreciate the request for computational details. The model was trained for a total of **57.9 hours on 16 H100 nodes**, each equipped with 8 GPUs. For inference, the **average time per frame is 23728 ms ± 207 ms**, measured on a single A6000 GPU. We will include these numbers in the final version to clarify the model’s efficiency and real-time applicability.
>
> ---
>
> ### **4. Main Failure Modes, Challenges in Long-Term Prediction, and Future Directions**
>
> The model tends to fail in scenarios involving **ambiguous egocentric motion**, such as rapid head turns with no clear landmarks or **asynchronous movement of body parts**. These cases challenge the model’s ability to temporally disambiguate motion and infer global structure from a first-person view.
>
> Moreover, **long-horizon degradation** is driven by error accumulation: small inaccuracies in early timesteps compound over time, leading to drift or overshooting. This is a common limitation of open-loop generative models. While some of this degradation is due to future uncertainty, it also reflects the lack of **corrective feedback** during inference.
>
> In future work, we plan to address this via training in **interactive environments** that allow feedback, or by incorporating **goal-conditioned** or **text-conditioned supervision** to help anchor predictions over extended rollouts.
>
> ---
>
> ### **5. Qualitative Visualization**
>
> Due to Program Chairs guidelines, we are unable to include generated videos in the rebuttal. However, we have included extensive visualizations in the supplementary material—such as sampled frames, temporal rollouts, and attention maps —to illustrate the model’s behavior. We hope these help convey the qualitative aspects of the generated sequences. We will include more visualizations of generated videos in the camera-ready version and we are happy to provide additional video results in an online repository upon request if there is a change in the conference guidelines.
>
> ---
>
> > \[1] Zhang, Siwei, et al. *Egobody: Human body shape and motion of interacting people from head-mounted devices*. European Conference on Computer Vision. Cham: Springer Nature Switzerland, 2022.

---

> > ### Author Response · Authors · 2025-08-04
> >
> > Dear Reviewer pyh8,
> >
> > Thank you again for your time and constructive reviews! With the discussion period drawing to a close, we expect your feedback and thoughts on our reply. We put a significant effort into our response, with several new experiments and discussions. We sincerely hope you can consider our reply in your assessment. We look forward to hearing from you, and we can further address unclear explanations and remaining concerns, if any.
> >
> > Best Regards,
> > Authors

---

> > > ### Comment · Reviewer_pyh8 · 2025-08-07
> > >
> > > Thank you for the clarification and additional experiments, which have addressed my concerns.

---

> > > > ### Author Response · Authors · 2025-08-07
> > > >
> > > > Dear Reviewer pyh8,
> > > >
> > > > Thank you again for taking the time to review our work and for your thoughtful feedback throughout the process. We're glad to hear that the clarification and additional experiments have addressed your concerns.
> > > >
> > > > If you feel that our responses have resolved the issues you raised, we would greatly appreciate it if you would consider updating your score accordingly. Of course, if there are any remaining concerns or questions, we would be more than happy to address them.
> > > >
> > > > We sincerely appreciate your contribution to the review process and your service to the community.
> > > >
> > > > Warm regards,
> > > > The Authors

---

### Official Review · Reviewer_t9bj · 2025-07-02

**Clarity:** 3
**Significance:** 2
**Originality:** 3
**Rating:** 4
**Confidence:** 3

**Summary:**

This paper proposes an interesting task that predicts future egocentric frames conditioned on human body trajectory. Specifically, conditioned on previous video frames and an action (representing human upper-body pose and root translation) for the current step, they trained a diffusion model to predict the video frame after executing the action. They showcased better results compared to baselines and also showed the potential of using the proposed model for planning tasks given a goal image.

**Questions:**

As described in the weakness part, there are some concerns about the current evaluation both quantitatively and qualitatively. In addition, it would be better to discuss whether 3-point poses (head+hand) are sufficient instead of full-body poses, clarify the confusion about the evaluation metrics and method description.

**Ethical Concerns:**

["NO or VERY MINOR ethics concerns only"]

**Final Justification:**

The rebuttal has provided more evaluations for hand pose accuracy and camera pose accuracy, and ablations of using 3-point input only. It addressed my major concern. I decide to raise my score.

**Limitations:**

yes

**Quality:**

3

**Strengths And Weaknesses:**

Strength:
- The task is interesting and has the potential to help with planning tasks.
- The paper showcased promising results of generating future video frames conditioned on past frames and action trajectories.
- They conducted various ablations to demonstrate the effectiveness of their proposed setting.
- They also showed some preliminary results of using the proposed model for action planning given a goal image.

Weakness:
There are some concerns about the current evaluation and presentation.
- The paper focused on using full-body pose as an action condition to generate future video frames. How accurate is the predicted future frame? Could there be some metrics for evaluating the accuracy of following actions beyond image space evaluation? For example, if it’s generating a frame with visible hand, maybe there could be a metric that evaluates the wrist position error in 2D (extract 2D keypoints from generated frame and compare with GT frame)? And maybe can also do camera pose estimation using the generated video frames and compare with GT camera pose (camera pose should be directly related to head pose)?
- Lack of more qualitative results. It would be better to have a supplementary video to showcase the generated video frames conditioned on the full-body action conditions (for example, show two videos simultaneously, left shows full-body human pose, right side shows the generated egocentric video).
- Another question is that for generating egocentric video, it seems head pose and hand pose are enough? Head pose is directly related to the camera pose of egocentric video, and thus is important to have. Hand poses are also important since sometimes the hands are visible in the egocentric videos. However, it’s not clear whether we need other joints’ poses for egocentric video generation. The current submission lacks discussion about this.
- Some descriptions and presentations in the paper are not clear. For example, in Table 2, what’s the evaluation metric here? In section 3.3, what does random timeskips mean?

---

> ### Author Rebuttal · Authors · 2025-07-31
>
> We sincerely thank the reviewer for their detailed and insightful review. Your feedback is incredibly valuable. Based on your review, we have conducted additional experiments for clarity. We discuss each point in depth below.
>
> ### **1. Evaluation Beyond Image Space: Measuring Action-Following Accuracy**
>
> We appreciate the reviewer’s suggestion to evaluate action-following accuracy beyond image-level metrics. In response, we introduce two additional evaluations:
>
> * **Wrist Position Evaluation**
>   We compute **PCK\@0.2**, precision, recall, and accuracy on 2D wrist keypoints across generated frames to measure local hand alignment.
>
> * **Pose Estimation Evaluation**
>   We estimate egocentric camera pose using **VGG-T** \[2] and compare it to ground truth using **ATE**, **RPE (Translation)**, and **RPE (Rotation)** \[1].
>
>
> These metrics directly evaluate whether the generated video faithfully reflects the conditioned action signal—both **locally** (e.g., visible hand positions) and **globally** (e.g., head-driven camera motion). **PEVA** outperforms all baselines, demonstrating its superior capacity for tracking embodied actions.
>
> #### **Wrist Position Evaluation across Models**
>
> | Model           | PCK\@0.2  | Accuracy  | Precision | Recall    |
> | --------------- | --------- | --------- | --------- | --------- |
> | DF              | 0.750     | 0.679     | 0.914     | 0.592     |
> | CDiT            | 0.755     | 0.794     | 0.865     | 0.833     |
> | **PEVA (Ours)** | **0.791** | **0.871** | **0.923** | **0.888** |
>
> #### **Camera Pose Evaluation across Models**
>
> | Model           | ATE ↓     | RPE (Translation) ↓ | RPE (Rotation) ↓ |
> | --------------- | --------- | ------------------- | ---------------- |
> | DF              | 0.464     | 0.422               | 36.514           |
> | CDiT            | 0.386     | 0.367               | 26.057           |
> | **PEVA (Ours)** | **0.274** | **0.266**           | **13.527**       |
>
> ---
>
> ### **2. Necessity of Whole-Body Conditioning**
>
> We thank the reviewer for raising this important question. To investigate whether head and hand pose alone are sufficient, we conduct an ablation study using only the 6-DoF poses of the head, left hand, and right hand (relative to their previous states).
>
> Although this subset captures visible body parts and head orientation, it leads to substantial degradation in both wrist and pose evaluations—highlighting the importance of full-body context.
>
> #### **Wrist Position Evaluation by Conditioning Type**
>
> | Conditioning  | PCK\@0.2  | Accuracy  | Precision | Recall    |
> | ------------- | --------- | --------- | --------- | --------- |
> | Head + Hands  | 0.692     | 0.756     | **0.906** | 0.722     |
> | **Full Body** | **0.877** | **0.858** | 0.890     | **0.907** |
>
> #### **Camera Pose Evaluation by Conditioning Type**
>
> | Conditioning  | ATE ↓     | RPE (Translation) ↓ | RPE (Rotation) ↓ |
> | ------------- | --------- | ------------------- | ---------------- |
> | Head + Hands  | 0.381     | 0.412               | 38.425           |
> | **Full Body** | **0.351** | **0.346**           | **26.578**       |
>
> These results indicate that **non-visible body joints**, such as those in the torso and legs, meaningfully affect egocentric visual dynamics—e.g., through head stabilization, balance, and intent. Full-body conditioning improves both **spatial grounding** and **long-term consistency**.
>
> ---
>
> ### **3. Lack of Qualitative Results / Videos**
>
> Due to Program Chairs guidelines, we are unable to include generated videos in the rebuttal. However, we have included extensive visualizations in the supplementary material—such as sampled frames, temporal rollouts, and attention maps —to illustrate the model’s behavior. We hope these help convey the qualitative aspects of the generated sequences. We will include more visualizations of generated videos in the camera-ready version and we are happy to provide additional video results in an online repository upon request if there is a change in the conference guidelines.
>
> ---
>
> ### **4. Clarity of Metrics and Terminology (e.g., Table 2 and Random Time Skips)**
>
> Thank you for this valuable feedback.
>
> * **Table 2**: The evaluation metric is **LPIPS** (Learned Perceptual Image Patch Similarity), which measures perceptual distance between generated and ground truth frames. We will clarify this in the table caption and main text.
>
> * **Random Time Skips**: Instead of training with a fixed timestep (e.g., every 0.5 s), we **randomly sample time intervals** during training and condition the model on the time delta. This improves generalization across short- and long-term predictions.
>
> To assess the effect of random time skips, we compare models trained at fixed frame rates (0.5 Hz, 1 Hz, 2 Hz, 4 Hz) against one trained with random time skips. We evaluate 1-step predictions at 1, 2, 4, 8, and 16 seconds using **DreamSim** perceptual loss.
>
> #### **Ablation: Random Time Skips vs Fixed Rate (↓ DreamSim)**
>
> | Method           | 1s        | 2s        | 4s        | 8s        | 16s       | Avg       |
> | ---------------- | --------- | --------- | --------- | --------- | --------- | --------- |
> | 4 Hz (fixed)     | 0.175     | 0.218     | 0.279     | 0.343     | 0.434     | 0.290     |
> | 2 Hz (fixed)     | 0.167     | 0.198     | 0.240     | 0.291     | 0.358     | 0.251     |
> | 0.5 Hz (fixed)   | 0.175     | 0.200     | 0.235     | 0.265     | 0.319     | 0.239     |
> | **Random Skips** | **0.169** | **0.198** | **0.236** | **0.268** | **0.293** | **0.233** |
>
> Fixed-rate models tend to overfit to specific temporal patterns and degrade at longer horizons. In contrast, **random time skips** encourage **temporally invariant learning**, improving both **short- and long-term prediction quality**.
>
> We will clarify this motivation and experiment in the revised paper.
>
> > \[1] Sturm et al. *"A benchmark for the evaluation of RGB-D SLAM systems."* IROS 2012
>
> > \[2] Wang et al. *"VGG-T: Visual Geometry Grounded Transformer."* CVPR 2025

---

> > ### Author Response · Authors · 2025-08-04
> >
> > Dear Reviewer t9bj,
> >
> > Thank you again for your time and constructive reviews! With the discussion period drawing to a close, we expect your feedback and thoughts on our reply. We put a significant effort into our response, with several new experiments and discussions. We sincerely hope you can consider our reply in your assessment. We look forward to hearing from you, and we can further address unclear explanations and remaining concerns, if any.
> >
> > Best Regards,
> > Authors

---

> > ### Comment · Reviewer_t9bj · 2025-08-07
> >
> > Thanks for providing more evaluations. It addressed my major concern. I will consider raising my score.

---

> > > ### Author Response · Authors · 2025-08-07
> > >
> > > Thank you — we're glad to hear that the additional evaluations addressed your concern. We appreciate your consideration and feedback!

---

### Official Review · Reviewer_6RQg · 2025-07-03

**Clarity:** 3
**Significance:** 3
**Originality:** 2
**Rating:** 5
**Confidence:** 2

**Summary:**

This work proposes a novel model for generating egocentric videos conditioned on a past video segment and a sequence of human actions. The approach introduces random time skips, sequence-level training, and action embeddings to ensure the model captures both short-term motion dynamics and long-term motion semantics. Experiments demonstrate that the proposed PEVA model generates high-quality egocentric videos with full-body motion control.

**Questions:**

See weaknesses

**Ethical Concerns:**

["NO or VERY MINOR ethics concerns only"]

**Final Justification:**

The rebuttal satisfactorily addressed my concerns about this paper. I would like to maintain my rating of accept.

**Limitations:**

This work presents a video diffusion model conditioned on motion. However, it does not account for actions involving the lower body (e.g., kicking a ball) or incorporate other control signals such as textual input.

**Quality:**

3

**Strengths And Weaknesses:**

Strengths

1. The paper is well-written and effectively presents the details of the proposed model. The ablation study clearly highlights the contribution of each design choice.
2. The experiments demonstrate that the model generates high-quality egocentric videos, outperforming baselines when conditioned on both atomic actions and long-term motion sequences.

Weaknesses

1. The proposed model builds upon existing diffusion-based methods by incorporating additional control signals, which limits the novelty of the contribution.
2. The role and importance of the random time skips in the model are not clearly justified in the ablation study.
3. There is no visualization of generated videos in the supplementary materials which would help illustrate the model.

---

> ### Author Rebuttal · Authors · 2025-07-31
>
> We thank the reviewer for the constructive feedback and for acknowledging the clarity of our model presentation and experiments. We appreciate the suggestions and have addressed the raised concerns as follows:
> ### **1. Novelty of the Contribution**
>
> In this work we address a new and underexplored problem: **how embodied physical actions causally shape first-person visual experience over time**. This problem is challenging, and so far has only been tackled with either simple action space (e.g., navigation) or in synthetic environments (e.g., computer games).
>
> To address this, we proposed **PEVA**, a novel autoregressive video model that generalizes to variable context lengths and can condition on physical actions. PEVA outperforms recent state-of-the-art action-conditioned diffusion models like **NWM** and **DF** across perceptual metrics such as **LPIPS** and **DreamSim**. Additionally, in a hand-controlled motion task, PEVA achieves **85% PCK\@0.1**, compared to only **22%** for the text-based model **Cosmos**, highlighting its ability to capture precise low-level control that language struggles to specify.
>
> Unlike prior diffusion-based methods, which typically use flat or semantically abstract conditioning (e.g., text, labels, global motion), PEVA leverages **hierarchically structured full-body kinematic trajectories**—aligned with the human joint graph. This enables visual prediction driven by specific embodied actions (e.g., head tilts, shoulder rolls, fine arm motions), and supports:
>
> * Disentangled action-perception analysis
> * Atomic counterfactuals
> * Goal-conditioned planning in pixel space
>
> The novelty lies not only in the modeling technique but in **proposing and solving a new class of controllable, embodied egocentric prediction problems**.
>
> ---
>
> ### **2. Justification for Random Time Skips**
>
> We thank the reviewer for highlighting the need for clarification.
>
> To evaluate the importance of random time skips, we conduct an ablation comparing models trained with fixed frame rates (**0.5 Hz, 1 Hz, 2 Hz, 4 Hz**) to our approach that was trained using **random time skips**. All models are evaluated on 1-step predictions at 1, 2, 4, 8, and 16 seconds using **DreamSim perceptual loss**.
>
> | **Method**            | **1s**    | **2s**    | **4s**    | **8s**    | **16s**   | **Avg**   |
> | --------------------- | --------- | --------- | --------- | --------- | --------- | --------- |
> | 4 Hz (fixed)          | 0.175     | 0.218     | 0.279     | 0.343     | 0.434     | 0.290     |
> | 2 Hz (fixed)          | 0.167     | 0.198     | 0.240     | 0.291     | 0.358     | 0.251     |
> | 0.5 Hz (fixed)        | 0.175     | 0.200     | 0.235     | 0.265     | 0.319     | 0.239     |
> | **Random Time Skips** | **0.169** | **0.198** | **0.236** | **0.268** | **0.293** | **0.233** |
>
> Fixed-rate training often **overfits to specific temporal dynamics** and degrades at longer horizons. In contrast, **random time skips encourage learning temporally invariant dynamics** and improve generalization across both short- and long-term predictions.
>
> We will clarify this motivation and experimental design more explicitly in the revised paper.
>
> ---
>
> ### **3. Visualization of Generated Videos**
>
> Due to Program Chairs guidelines, we are unable to include generated videos in the rebuttal. However, we have included **extensive visualizations in the supplementary material**—such as sampled frames, temporal rollouts, and attention maps—to illustrate the model’s behavior.
>
> We hope these help convey the qualitative aspects of the generated sequences. We will include more visualizations of generated videos in the **camera-ready version**, and we are happy to provide additional video results in an **online repository** upon request if there is a change in the conference guidelines.

---

> > ### Author Response · Authors · 2025-08-04
> >
> > Dear Reviewer 6RQg,
> >
> > Thank you again for your time and constructive reviews! With the discussion period drawing to a close, we expect your feedback and thoughts on our reply. We put a significant effort into our response, with several new experiments and discussions. We sincerely hope you can consider our reply in your assessment. We look forward to hearing from you, and we can further address unclear explanations and remaining concerns, if any.
> >
> > Best Regards,
> > Authors

---

### Comment · Area_Chair_4DrJ · 2025-08-06
**Rebuttal posted, please engage in discussion!**

Hi Reviewers,

The author has posted their rebuttal. Could you please review their response and share your thoughts?

Please engage in the discussion as soon as possible to allow for a meaningful back-and-forth before the deadline.

Thank you for your timely attention to this.

Best,

Your AC

---

### Decision · Program_Chairs · 2025-09-17

**Decision:**

Accept (poster)

**Comment:**

This paper introduces PEVA, a model designed to predict future ego-centric video conditioned on past video and the human agent's actions, which are represented by relative 3D body pose trajectories. The authors also contribute a hierarchical evaluation protocol to systematically assess the model's capabilities in embodied prediction and control.

The initial reviews for this paper were mixed, with some reviewers raising points that required further clarification. However, the authors provided a comprehensive rebuttal that effectively addressed the concerns. Following the author's response and subsequent discussion, all reviewers reached a consensus and are now in favor of accepting the paper.

I concur with the reviewers' final recommendation. This is an interesting and well-executed work that makes a valuable contribution toward the ambitious goal of creating ego-centric world models. The proposed method is sound and will likely benefit the community. The authors are encouraged to revise the final version of their manuscript to incorporate the suggestions raised by the reviewers and reflect the valuable points from the discussion period.